# A novel haemocytometric COVID-19 prognostic score developed and validated in an observational multicentre European hospital-based study

Joachim Linssen[1], Anthony Ermens[2], Marvin Berrevoets[3], Michela Seghezzi[4], Giulia Previtali[4], Simone van der Sar-van der Brugge[2], Henk Russcher[5], Annelies Verbon[5], Judith Gillis[6], Jürgen Riedl[7], Eva de Jongh[7], Jarob Saker[1], Marion Münster[1], Imke CA Munnix[8], Anthonius Dofferhof[8], Volkher Scharnhorst[9], Heidi Ammerlaan[9], Kathleen Deiteren[10], Stephan JL Bakker[11], Lucas Joost Van Pelt[11], Yvette Kluiters-de Hingh[3], Mathie PG Leers[12], Andre J van der Ven[13]*

[1]Sysmex Europe GmbH, Hamburg, Germany; [2]Amphia Hospital, Breda, Netherlands; [3]Elisabeth-Tweesteden Hospital, Tilburg, Netherlands; [4]Hospital Papa Giovanni XXIII, Bergamo, Italy; [5]Erasmus MC, University Medical Center, Rotterdam, Netherlands; [6]Leiden University Medical Center, Leiden, Netherlands; [7]Albert Schweitzer Hospital, Dordrecht, Netherlands; [8]Canisius Wilhelmina Hospital, Nijmegen, Netherlands; [9]Catharina Hospital, Eindhoven, Netherlands; [10]University Hospital Antwerp, Antwerp, Belgium; [11]University Medical Center Groningen, University of Groningen, Groningen, Netherlands; [12]Zuyderland Medical Center, Sittard-Geleen, Netherlands; [13]Radboud University Medical Center, Nijmegen, Netherlands

*For correspondence:
andre.vanderven@radboudumc.nl

**Abstract** COVID-19 induces haemocytometric changes. Complete blood count changes, including new cell activation parameters, from 982 confirmed COVID-19 adult patients from 11 European hospitals were retrospectively analysed for distinctive patterns based on age, gender, clinical severity, symptom duration, and hospital days. The observed haemocytometric patterns formed the basis to develop a multi-haemocytometric-parameter prognostic score to predict, during the first three days after presentation, which patients will recover without ventilation or deteriorate within a two-week timeframe, needing intensive care or with fatal outcome. The prognostic score, with ROC curve AUC at baseline of 0.753 (95% CI 0.723–0.781) increasing to 0.875 (95% CI 0.806–0.926) on day 3, was superior to any individual parameter at distinguishing between clinical severity. Findings were confirmed in a validation cohort. Aim is that the score and haemocytometry results are simultaneously provided by analyser software, enabling wide applicability of the score as haemocytometry is commonly requested in COVID-19 patients.

## Introduction

COVID-19 spans a wide clinical spectrum from asymptomatic to severe pneumonia with multiple organ failure (*Guan et al., 2020*), majorly threatening global health, including that of Europe (*Sun et al., 2020a*). Early identification of critical patients may reduce mortality by timely interventions (*Sun et al., 2020a*). Many studies have explored the diagnostic or prognostic value of various factors including age, sex, CT scan, biochemical, and haematological parameters (*Galloway et al.,*

*2020*; *Ji et al., 2020*; *Lu et al., 2020*; *Shi et al., 2020a*; *Shi et al., 2020b*; *Sun et al., 2020b*; *Wang et al., 2020a*; *Yan et al., 2020*; *Caramelo et al., 2020*; *Yuan et al., 2020*; *Chen et al., 2020*). Most studies were however geographically limited, had high risk for bias, and had no validation cohort (*Wynants et al., 2020*). C-reactive protein, ferritin, D-dimer, albumin, urea nitrogen, bilirubin, and lactate dehydrogenase (LDH) levels are cited as indirect indicators of the presence and severity of COVID-19 (*Henry et al., 2020*; *Ji et al., 2020*; *Liu et al., 2020a*; *Luo et al., 2020*; *Sun et al., 2020b*; *Wu et al., 2020*; *Yang et al., 2020a*; *Zhou et al., 2020*), as are complete blood count (CBC) and differential count (DIFF) changes, specifically lymphopenia, neutrophilia, high neutrophil-to-lymphocyte ratio (NLR) and thrombocytopenia (*Henry et al., 2020*; *Jiang et al., 2020*; *Kermali et al., 2020*; *Khartabil et al., 2020*; *Lippi et al., 2020a*; *Lippi and Plebani, 2020b*). All aforementioned parameters are widely available, but their value is constrained by significant inter-patient variability and limited specificity.

Newer haematology analysers are capable of functional characterisation of blood cells (*Hoffmann, 2014*; *Buttarello, 2016*; *Chabot-Richards and George, 2015*), including measurement of immune cell activation (*Kono et al., 2018*; *Linssen et al., 2007*), which has shown promise in screening for infectious diseases (*Prodjosoewojo et al., 2019*). The aim of this study is to develop and validate a prognostic score using only haemocytometric data for COVID-19 patients presenting at hospitals, to predict within three days of hospital admission, who will deteriorate and require intensive care unit (ICU) transfer within 14 days of admission. Importantly, our intended purpose of this score is to assist with objective risk stratification to support patient management decision making early on, and thus facilitate timely interventions, such as need for ICU or not, before symptoms of severe illness become clinically overt, with the intention to improve patient outcomes, and not to predict mortality. A secondary aim of this study is to document trends of haematology parameters over time, specifically for the newer parameters, as most of the published data in COVID-19 patients focussed on traditional parameters such as lymphocyte counts, platelet counts, and NLR.

## Results

### Patient characteristics

In total 999 patients were enrolled in the development cohort (*Figure 1*). Seventeen patients with underlying haematological malignancies or currently undergoing chemotherapy, were subsequently excluded. Nine hundred eighty-two patients with 2587 haematology measurements (day 0–13), were included to analyse and document temporal haemocytometric data trends. Median age was 71 years (range 18-96) and 68% of the patients were male. Patient distribution by sex, clinical severity, and comorbidities is shown in *Table 1* with a breakdown by hospital shown as *Supplementary file 1*. After excluding 59 patients with missing day 0–3 CBC-Diff data, the remaining 923 patients with 1587 measurements were used for the prognostic score development.

Of the 923 patients, 64 (6.9%) were not hospitalised, 687 (74.4%) were admitted to general wards and 172 (18.6%) went directly to ICU, with a further 9 ICU transfers after day 3. The mortality rate for ICU patients (29.3%; 53/181) and general ward patients (27.4%, 186/678) was comparable (*Figure 1*).

Patients who died or were critically ill, were significantly older than those less severely ill (median age 74 vs 65 years, p<0.001). Although males outnumbered females (631; 292) in patients that had a severe disease progression, mortality rates were independent of sex. Characteristics of the 923 patients are presented in *Table 2*. Distribution of clinical severity by age is shown in *Figure 2*.

### Haemocytometry data trends over 14 days of hospitalisation in critical illness (CI) and non-critical (NC) patients

The haematological changes derived from the 2587 measurements, taken from 982 patients at various time points (at the discretion of the attending physician) from day of admission (day 0) up to 13 days, are shown in *Figure 3* (lymphocyte-related parameters), *Figure 4* (neutrophil-related parameters), *Figure 5* (monocyte parameters), *Figure 6* (red blood cell-related parameters), and *Figure 7* (platelet parameters) and further described below with findings grouped along haemopoietic cell lines.

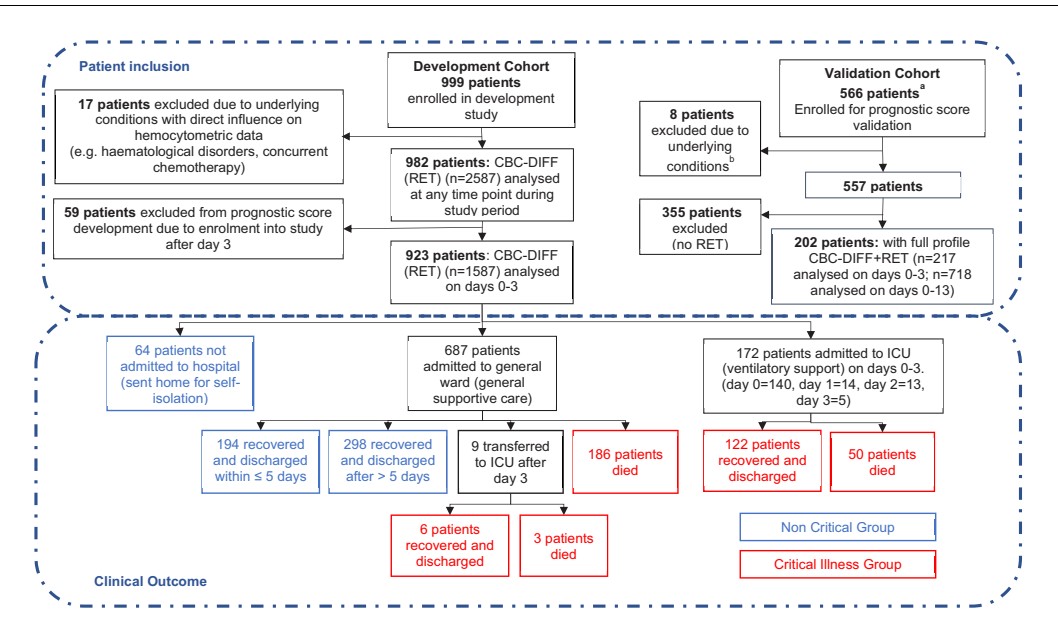

**Figure 1.** Patient flow chart from prognostic score development and validation cohorts, including sample numbers for CBC-Diff (with or without RET) and clinical outcome. Footnote. (a) Details of how the validation patient cohort patients were selected are provided in *Figure 10*, (b) the exclusion criteria for the validation cohort were the same as for the development patient cohort.

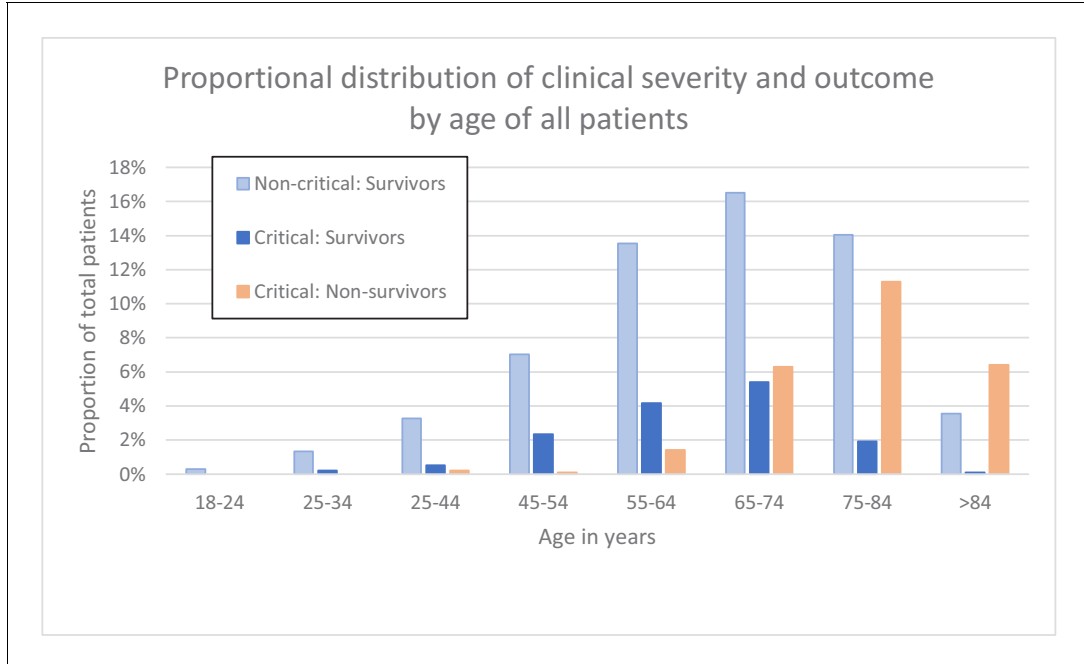

**Figure 2.** Clinical severity and outcome by age of all patients.

The online version of this article includes the following source data for figure 2:

**Source data 1.** Source data for clinical severity and outcome by age of all patients.

**Table 1.** Basic demographic characteristics of COVID-19 PCR confirmed patients enrolled for prognostic score development.

| | Total | Mild | Moderate | Severe | Critical | Fatal |
|---|---|---|---|---|---|---|
| All patients n [%] | 982 | 64 [6.5] | 198 [20.2] | 323 [32.9] | 144 [14.7] | 253 [25.8] |
| Age Years (range) | 18–96 | 18–96 | 22–91 | 19–93 | 28–88 | 42–95 |
| Age Years (median) | 71 | 59 | 68 | 69,5 | 65 | 79 |
| Females | | | | | | |
| n [%] | 314 [32.0] | 27 [8.6] | 74 [23.6] | 113 [36.0] | 38 [12.1] | 62 [19.7] |
| Age Years (range) | 18–95 | 18–92 | 22–86 | 19–93 | 28–81 | 56–95 |
| Age Years (median) | 69 | 62 | 67 | 69 | 64 | 79.5 |
| Males | | | | | | |
| n, [%] | 668 [68.0] | 37 [5.5] | 124 [18.6] | 210 [31.4] | 106 [15.9] | 191 [28.6] |
| Age Years (range) | 26–96 | 35–96 | 30–91 | 26–89 | 32–88 | 42–93 |
| Age Years (median) | 72 | 59 | 68 | 69.5 | 65 | 79 |
| Length of hospitalisation | | | | | | |
| Days (range) | 0–78 | 0 | 1–5 | 6–46 | 2–78 | 1–44 |
| Days (median) | 6 | 0 | 3 | 10 | 22 | 6 |
| Comorbidities* | | | | | | |
| Absent† [%] | [27] | [68] | [45] | [35] | [24] | [13] |
| Present‡ [%] | [73] | [32] | [55] | [65] | [76] | [87] |
| Diabetes [%] | [15.9] | [5.5] | [15.0] | [14.9] | [20.5] | [18.2] |
| Hypertension [%] | [12.8] | [1.8] | [7.9] | [14.5] | [17.9] | [13.9] |
| Cardiovascular disorders [%] | [27.6] | [16.3] | [21.3] | [25.6] | [18.8] | [41.6] |
| Respiratory disorders [%] | [12.0] | [5.5] | [8.7] | [12.6] | [12.5] | [14.8] |
| CNS disorders [%] | [4.3] | [3.6] | [3.2] | [4.6] | [1.8] | [6.2] |
| Renal disorders [%] | [4.8] | [1.8] | [3.9] | [5.0] | [5.4] | [5.7] |
| Malignancy [%] | [9.2] | [0] | [9.5] | [8.4] | [9.8] | [12.0] |
| Autoimmune disease [%] | [3.7] | [5.5] | [3.9] | [2.3] | [6.3] | [3.3] |
| Pregnancy [%] | [0.3] | [1.8] | [0.8] | [0] | [0] | [0] |
| Obesity§ [%] | [2.7] | [0] | [3.1] | [3.4] | [1.8] | [2.9] |
| Other¶ [%] | [7.8] | [5.5] | [4.7] | [10.7] | [12.5] | [4.3] |

\* Occurrence of comorbidities is shown as a relative frequency expressed as a percentage (patients with a comorbidity divided by total number of patients in whom the presence or absence of comorbidities was recorded) as 2 of the participating study centres did not document the presence or absence of comorbidities.

† Absent – comorbidities confirmed to be absent.

‡ Present – one or more comorbidities confirmed to be present.

§ Body mass index (BMI) measurements were not undertaken as part of this study. It was also up to the discretion of the physician to document obesity as a comorbidity. As this study was undertaken in the early phase of the pandemic, it may not have been common knowledge to all attending physicians that obesity is significant contributor to adverse outcomes in COVID-19.

¶ Other includes gastrointestinal disorders, musculoskeletal disorders, endocrine disorders, lipid disorders, haemochromatosis psoriasis, malnutrition, and Down syndrome.

The online version of this article includes the following source data for  Table 1:

Source data 1. Source data for basic demographic characteristics of COVID-19 PCR confirmed patients enrolled for prognostic score development.

**Table 2.** Demographic and first haemocytometric data at hospital presentation or within 3 days after admission.

| | Non-critical group (NC) | Critical illness group (CI) | P value | All patients | Reference values healthy volunteers | % abnormal results |
|---|---|---|---|---|---|---|
| Patients [n] | 557 | 366 | <0.0001 | 923 | | |
| Female [n] | 201 | 91 | <0.0001 | 292 | | |
| Male [n] | 356 | 276 | <0.0001 | 631 | | |
| Age (Years, median) | 65 | 74 | <0.0001 | 68 | | |
| Female [Y, median] | 64 | 73 | 0.0040 | 67 | | |
| Male [Y, median] | 66 | 74 | 0.0010 | 69 | | |
| Female vs. Male in NC [n] [F, M] | [201, 356] | na | ns | na | | |
| Female vs. Male in CI [n] [F, M] | na | [91, 276] | ns | na | | |
| Duration of symptoms [Days, median] | 7 | 7 | ns | 7 | | |
| Haemocytometry | | | | | | |
| Patients/healthy volunteers, [samples] | 557, [899] | 366, [688] | | 923, [1587] | 12782 | |
| White blood cells findings | Median [95% CI] | Median [95% CI] | P value | Median [95% CI] | Median [95% CI] | % |
| WBC (white blood cell count) [$10^3$/µL] | 5.92 [2.42–13.17] | 7.75 [3.02–18.61] | <0.0001 | 6.43 [2.59–16.10] | 5.84 [3.64–9.61] | 28.4 |
| NEUT (neutrophil count) [$10^3$/µL] | 4.37 [1.39–11.21] | 6.25 [2.17–15.18] | <0.0001 | 4.96 [1.53–13.61] | 3.12 [1.62–5.86] | 41.4 |
| IG (immature granulocyte count) [$10^3$/µL] | 0.04 [0.01–0.26] | 0.06 [0.01–0.67] | <0.0001 | 0.05 [0.01–0.48] | 0.03 [0.01–0.09] | 16.3 |
| LYMPH (lymphocyte count) [$10^3$/µL] | 0.91 [0.35–2.09] | 0.74 [0.26–1.91] | <0.0001 | 0.78 [0.28–2.00] | 1.92 [1.07–3.41] | 70.5 |
| NLR (neutrophil-to-lymphocyte ratio) [ratio] | 4.7 [1.3–17.7] | 8.1 [2.1–39.0] | <0.0001 | 6.0 [1.4–28.5] | 1.6 [0.8–3.6] | 77.1 |
| IGLR (immature granulocyte-to-lymphocyte ratio) [ratio*100] | 3.9 [0.8–36.1] | 7.3 [1.2–69.1] | <0.0001 | 4.8 [0.81–52.7] | 1.8 [0.5–4.8] | 49.4 |
| MONO (monocyte count) [$10^3$/µL] | 0.43 [0.13–1.29] | 0.43 [0.10–1.22] | 0.0172 | 0.42 [0.11–1.27] | 0.48 [0.28–0.83] | 22.8 |
| EO (eosinophil count) [$10^3$/µL] | 0.01 [0.00–0.26] | 0.01 [0.00–0.31] | ns. | 0.01 [0.00–0.29] | 0.16 [0.04–0.47] | 76.6 |
| BASO (basophil count) [$10^3$/µL] | 0.01 [0.00–0.05] | 0.01 [0.00–0.08] | ns. | 0.01 [0.00–0.07] | 0.04 [0.02–0.09] | 56.1 |
| WBC extended parameters | | | | | | |
| NEUT-RI (neutrophil reactivity index) [FI] | 50.0 [43.9–57.0] | 50.6 [44.2–59.2] | 0.0023 | 50.2 [44.0–58.1] | 46.1 [41.9–50.6] | 47.0 |
| NEUT-GI (neutrophil granularity index) [GI] | 153 [143–163] | 153 [141–163] | ns | 153 [143–163] | 149 [142–157] | 23.2 |
| RE-LYMPH (reactive lymphocytes as % of lymphocytes) [% of lymph] | 9.9 [2.6–23.6] | 10.0 [2.4–28.9] | <0.0001 | 10.5 [2.5–26.3] | 3.3 [1.3–7.6] | 73.3 |
| AS-LYMPH (antibody-synthesising lymphocytes as % of lymphocytes) [% of lymph] | 3.6 [0.0–13.2] | 4.2 [0.0–19.8] | <0.0001 | 3.8 [0.0–17.4] | 0.0 [0.0–0.0] | 90.4 |
| RE-MONO (reactive monocytes as % of monocytes) [% of mono] | 2.5 [0.2–17.9] | 3.9 [0.0–30.0] | <0.0001 | 3.0 [0.3–22.7] | 0.0 [0.0–4.3] | 37.3 |
| Red blood cells findings | | | | | | |
| HGB (haemoglobin) [g/dL] | 13.4 [9.6–16.3] | 12.9 [8.4–16.5] | <0.0001 | 13.2 [8.9–16.3] | 14.1 [11.9–16.6] | 26.5 |
| RBC (red blood cell count) [$10^6$/µL] | 4.8 [3.10–5.54] | 4.31 [2.90–5.62] | <0.0001 | 4.42 [3.00–5.58] | 4,73 [3.99–5.60] | 26.8 |
| MCV (mean cell volume) [fL] | 89.1 [78.3–101.6] | 90.2 [77.9–103.0] | ns | 89.5 [78.3–101.9] | 90.1 [82.5–97.8] | 14.2 |
| RET (reticulocyte count) [$10^3$/µL] | 30.2 [14.1–80.0] | 30.0 [13.9–95.6] | ns | 30.1 [14.1–89.5] | 57.5 [32.6–96.6] | 55.5 |
| NRBC (nucleated red blood cell count) [/µL] | 0 [0–10] | 0 [0–60] | <0.0001 | 0 [0–30] | 0 [0–10] | 46 |

*Table 2 continued on next page*

*Table 2 continued*

| | Non-critical group (NC) | Critical illness group (CI) | P value | All patients | Reference values healthy volunteers | % abnormal results |
|---|---|---|---|---|---|---|
| RET-He (reticulocyte haemoglobin equivalent) [pg] | 30.2 [23.6–35.5] | 29.4 [22.9–34.8] | ns | 29.9 [23.1–35.2] | 32.8 [29.4–35.4] | 44.4 |
| RBC extended parameters | | | | | | |
| DELTA-He (difference in reticulocyte and RBC haemoglobin content) [pg] | −0.5 [−4.8–3.1] | −1.2 [−7.0–3.1] | <0.0001 | −0.7 [−5.9–3.1] | 2.6 [1.4–3.6] | 85.7 |
| MICRO-R (% of RBC that are microcytic) [%] | 1.5 [0.4–8.5] | 1.5 [0.4–10.0] | ns | 1.5 [0.4–9.5] | 1.1 [0.3–4.2] | 9.9 |
| HYPO-He (% of RBC that are hypochromic) [%] | 0.2 [0.1–2.0] | 0.3 [0.1–2.6] | <0.0001 | 0.2 [0.1–2.5] | 0.1 [0.0–0.5] | 21.9 |
| Platelet findings | | | | | | |
| PLT (platelet count) [$10^3$/μL] | 199 [96–451] | 205 [64–435] | ns | 201 [79–446] | 256 [161–385] | 32.7 |
| IPF% (immature platelet fraction) [%] | 4.3 [1.7–12.9] | 4.9 [2.0–5.9] | ns | 4.5 [1.7–14.2] | 3.0 [1.1–8.1] | 14.6 |
| PLT extended parameters | | | | | | |
| IPF# (absolute immature platelet count) [$10^3$/μL] | 9.4 [3.6–30.9] | 11.2 [3.5–28.7] | 0.0001 | 10.2 [3.5–29.6] | 7.9 [3.1–18.1] | 15.4 |

Note: The expanded name of the haemocytometric parameters is shown in brackets in italics. The unit of measure for each parameter is shown in square brackets. A more detailed explanation of the haemocytometric parameters is provided in *Table 7* .

Abbreviations: na, not applicable; ns, not significant.

The online version of this article includes the following source data for Table 2:

Source data 1. Source data for demographic and first haemocytometric data available for patients at hospital presentation or within 3 days after admission.

## WBC findings

In NC and CI groups, lymphopenia is present for 7 and 10 days respectively and normalises thereafter, as lymphocyte numbers tend to increase after 5 days in both groups. (*Figure 3A*) The NLR increases in the CI group compared to the NC group, and then gradually decreases again. The differences between the groups remain significant over time (p<0.001) (*Figure 3B*). Neutrophil counts are normal and remain stable in the NC group, whereas values are mildly elevated and continue to rise over time in the CI group (*Figure 4A*). This increase in neutrophils is accompanied by a mildly elevated NEUT-RI level in the CI group (*Figure 4B*). Absolute differences in baseline IG, although statistically significant (p<0.001), are small. After day 2, there is a marked rise in IG in the CI but not in the NC group (*Figure 4C*). The immature granulocyte-to-lymphocyte ratio (IGLR) trend mirrors that of IG (*Figure 4D*). AS-LYMPH%/L remains abnormally elevated in both groups from baseline throughout the first two weeks of hospitalisation (*Figure 3C*). The RE-LYMPH, minus AS-LYMPH (a subset of RE-LYMPH, which is absent in healthy individuals) indicates that the reactive lymphocytes observed in both NC and CI groups are largely comprised of AS-LYMPH (*Figure 3D*), in keeping with a predominant B-cell rather than T-cell response. Absolute monocyte counts are within normal limits for the NC group and remain stable throughout. Likewise, values are largely normal within the CI group, but show an upward trend with mild monocytosis evident from about day 10 onwards (*Figure 5A*). RE-MONO%/M remains stable over time, and largely within normal limits in the NC group. In contrast, in the CI group monocyte activation increases up to days 3 to 4, with values returning to within the reference range after a week (*Figure 5B*).

## RBC findings

In all patients, there is a gradual drop in HGB, also after adjusting for age and sex (*Figure 6A*), with differences between the groups becoming increasingly wider from day 5 onwards. After day 7, HGB continues to decline only in the CI group. RET# remain low in both groups despite dropping HGB in the first week but RET# shows a consistent rise thereafter in the CI group, towards the upper limit of the normal reference range (*Figure 6B*).

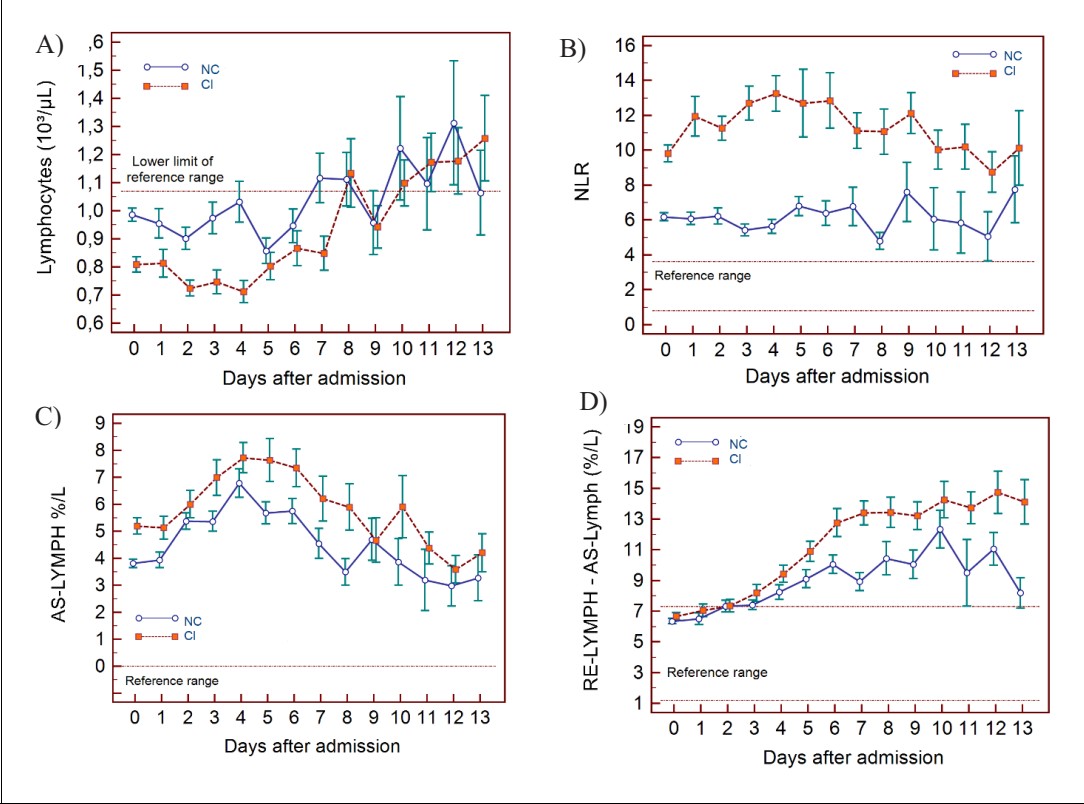

**Figure 3.** Trends of lymphocyte-related parameters over 14 days of hospitalisation in critical illness (CI) and non-critical (NC) patients. Note: 14 days of hospitalisation refers to Day 0 (day of admission) plus the next 13 days after admission. The normal reference range is depicted by the area between the dotted horizontal lines. Vertical bars indicate standard error of the mean (SEM). (A) Absolute lymphocyte count (LYMPH), (B) neutrophil-to-lymphocyte ratio (NLR), (C) antibody-synthesising lymphocytes as percentage of lymphocytes (AS-LYMPH%/L), (D) reactive lymphocytes minus AS-LYMPH (as a percentage of lymphocytes). The number of sample measurements available per day for the trend analysis for the parameters plotted per patient group are shown in *Figure 3—source data 1*.

The online version of this article includes the following source data for figure 3:

**Source data 1.** Source data for number of measurements for each day of hospitalisation that were available per patient group for the trends of lymphocyte-related parameters over 14 days of hospitalisation.

**Source data 2.** Source data for trends of absolute lymphocyte count over 14 days of hospitalisation in critical illness (CI) and non-critical (NC) patients.

**Source data 3.** Source data for trends of neutrophil-to-lymphocyte ratio over 14 days of hospitalisation in critical illness (CI) and non-critical (NC) patients.

**Source data 4.** Source data for trends of AS-LYMPH%/L over 14 days of hospitalisation in critical illness (CI) and non-critical (NC) patients.

**Source data 5.** Source data for trends of RE-Lymph minus AS-LYMPH as a percentage of total lymphocytes over 14 days of hospitalisation in critical illness (CI) and non-critical (NC) patients.

The Delta-He is negative and remains relatively stable in the NC group (*Figure 6C*). In contrast, Delta-He drops progressively in the CI group, reaching its nadir at about day 7, and then rises towards zero, primarily due to an improvement in the RET-He values (data not shown). NRBC# is almost zero in the NC group (within normal range) but rises sharply and progressively at about day 5 in the CI group (*Figure 6D*).

## PLT findings

PLT are largely within the normal range but show a progressive upward trend over time for both groups, with patients in the CI group manifesting with mild thrombocytosis from about day 10 onwards (*Figure 7A*). The IPF# initially is within the normal reference range for both groups but over time the CI group shows a gradual increase, exceeding the upper limit of the reference range in parallel to PLT (*Figure 7C*), whereas IPF(%) remains within normal limits throughout for both groups (*Figure 7D*). The platelet-to-lymphocyte ratio (PLR) is abnormally elevated for both groups

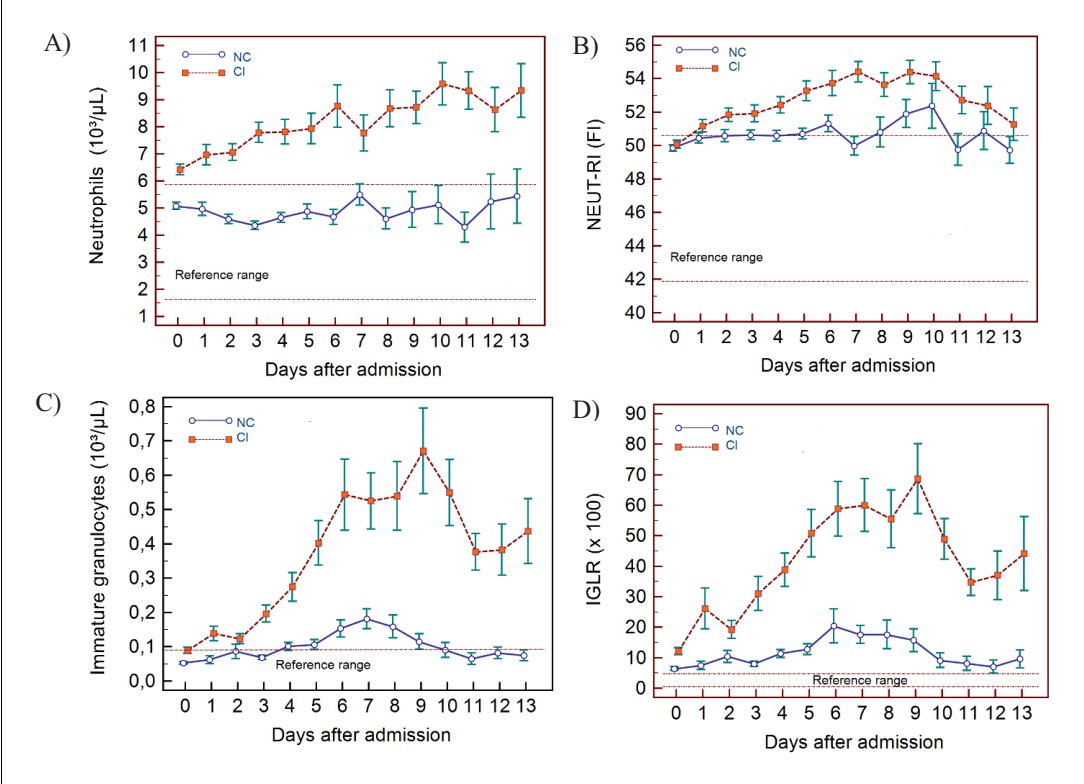

**Figure 4.** Trends of neutrophil-related parameters over 14 days of hospitalisation in critical illness (CI) and non-critical (NC) patients. Note: 14 days refers to day 0 (day of admission) plus the next 13 days after admission. The normal reference range is depicted by the area between the dotted horizontal lines. Vertical bars indicate standard error of the mean (SEM). (A) Absolute neutrophil count (NEUT), (B) neutrophil reactivity index (NEUT-RI), (C) immature granulocytes (IG), (D) Immature granulocyte-to-lymphocyte ratio *100 (IGLR). The number of sample measurements available per day for the trend analysis for the parameters plotted per patient group are shown in *Figure 4—source data 1*.

The online version of this article includes the following source data for figure 4:

**Source data 1.** Source data for number of measurements for each day of hospitalisation that were available per patient group for the trends of neutrophil-related parameters over 14 days of hospitalisation.

**Source data 2.** Source data for trends of absolute neutrophil count over 14 days of hospitalisation in critical illness (CI) and non-critical (NC) patients.

**Source data 3.** Source data for trends of NEUT-RI over 14 days of hospitalisation in critical illness (CI) and non-critical (NC) patients.

**Source data 4.** Source data for trends of immature granulocyte count over 14 days of hospitalisation in critical illness (CI) and non-critical (NC) patients.

**Source data 5.** Source data for trends of immature granuclocyte-to-lymphocyte ratio over 14 days of hospitalisation in critical illness (CI) and non-critical (NC) patients.

throughout, with values slightly higher in the CI group, but only until day 5, after which the NC and CI groups overlap (*Figure 7B*).

## Formulation and performance of the haemocytometric COVID-19 prognostic score in the development cohort

In the analysis of 1587 samples, 923 patients (days 0-3), six variables (parameters or ratios thereof) were identified that fulfilled the prespecified selection criteria, namely sufficient discriminatory power between NC and CI patient groups ($p \leq 0.001$) and at least 20% of all CI results outside of the normal range. These were NLR ($p<0.0001$, 77.1%), IGLR ($p<0.0001$, 49,4%), RE-MONO%/M ($p<0.0001$, 37.3%), AS-LYMPH%/L ($p<0.0001$, 90,4%), Delta-He ($p<0.0001$, 85,7%), and NRBC ($p<0.0001$, 46.0%). Other parameters also fulfilled these criteria, but these were not selected as they are interdependent on others already included. These were WBC, NEUT#, LYMPH# (represented by NLR, IGLR, AS-LYMPH), MONO# (represented by RE-MONO), RE-LYMPH (represented by AS-LYMPH), RBC, HGB, and HYPO-He (represented by DELTA-He) as well as NEUT-RI which is commonly elevated in bacterial infection (*Prodjosoewojo et al., 2019*).

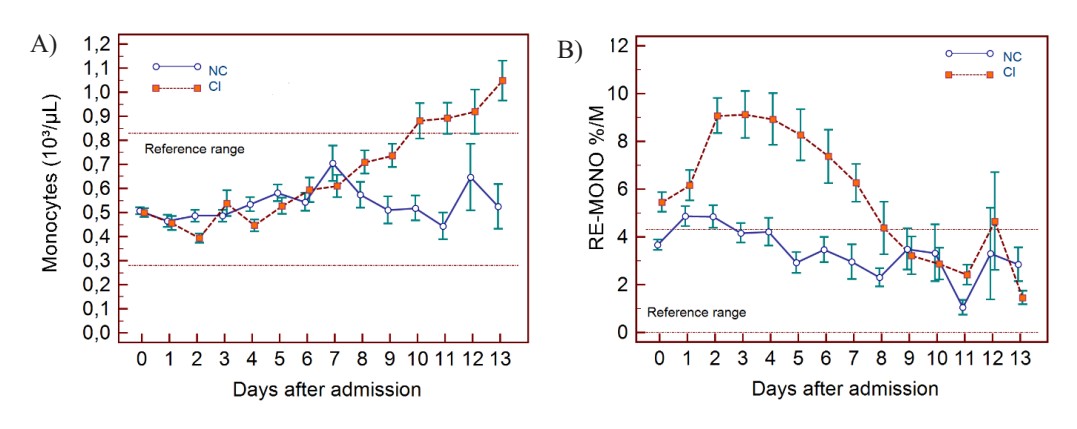

**Figure 5.** Trends of monocyte parameters over 14 days of hospitalisation in critical illness (CI) and non-critical (NC) patients. Note: 14 days of hospitalisation refers to day 0 (day of admission) plus the first 13 days after admission. The normal reference range is depicted by the area between the dotted horizontal lines. Vertical bars indicate standard error of the mean (SEM). (A) Absolute monocyte count (MONO), (B) reactive monocytes as a percentage of monocytes (RE-MONO%/M). The number of sample measurements available per day for the trend analysis for the parameters plotted per patient group are shown in *Figure 5—source data 1*.

The online version of this article includes the following source data for figure 5:

**Source data 1.** Source data for number of measurements for each day of hospitalisation that were available per patient group for the trends of monocyte parameters over 14 days of hospitalisation.

**Source data 2.** Source data for trends of absolute monocyte count over 14 days of hospitalisation in critical illness (CI) and non-critical (NC) patients.

**Source data 3.** Source data of trends of reactive monocytes as a percentage of total monocyte count over 14 days of hospitalisation in critical illness (CI) and non-critical (NC) patients.

These six variables (NLR, IGLR, RE-MONO%/M, AS-LYMPH%/L, Delta-He, NRBC) were assigned a score from 0 to maximum of 4 points each. Applying this score to the CI and NC patient groups, no overlap (SEM) was noticed in the mean values except on day 13, where the available data points were few (*Figure 8A*). These combined six variables had a sensitivity of 68% (249/366) in identifying CI patients in the first 3 days. However, clear haematological abnormalities, especially HGB, HYPO-He, PLT, and IPF# were observed in 8 of the undetected CI patients. These parameters were therefore added to the score with a maximum of 1 point each. The sum of 10 variables makes up the final prognostic score value, with a minimum of 0 and a theoretical maximum 28 points. The prognostic score cut-off values and point allocation matrix is shown in *Table 3*.

Prognostic score performance in the 923 patients using 3 as the cut-off, correctly identified 70.5% (95% CI 66–75) of patients finally classified as critical illness (CI group) on days 0–3 with sensitivities of 62.3% (95% CI 55–69), 74.5% (95% CI 61–85), 75.6% (95% CI 64–85), and 87.0% (95% CI 77–94) on days 0, 1, 2, and 3 respectively, and 93% (95% CI 81-99) on day 6. Moreover, all patients classified with critical illness, whether they recovered or died, had an increasing score value, compared to those that recovered without ICU intervention (*Figure 8B*). Specifically, the scores for non-ICU patients who subsequently deteriorated and died (n=186, *Figure 1*), were already notably higher during the initial phase of hospitalisation with a median value of 3 (95% CI 2-4), 4 (95% CI 1.5-6.7), 5 (95% CI 4.0-6.0), and 5 (95% CI 4.4-7.6) on days 0, 1, 2, and 3 respectively, (*Table 4*), compared with 1 (95% CI 1.0-2.0) to 2 (95% CI 1.0-3.0) for patients who recovered without ICU intervention.

In patients classified as having 'critical illness', irrespective of whether the final outcome was death (with or without ICU intervention) or recovery, the prognostic score values increased progressively over time with no statistically significant difference (p>0.05) in prognostic score values between survivors and non-survivors, confirming that our score predicts a critical clinical course, but not the outcome, that is recovery or death.

For the initial period of hospitalisation (<4 days) and for the 14-day prediction time horizon, the prognostic score was better than NLR at differentiating clinical severity, with a higher AUC at all time points (*Table 5*). Notably, the cut-off value that determined the best AUC was consistent for

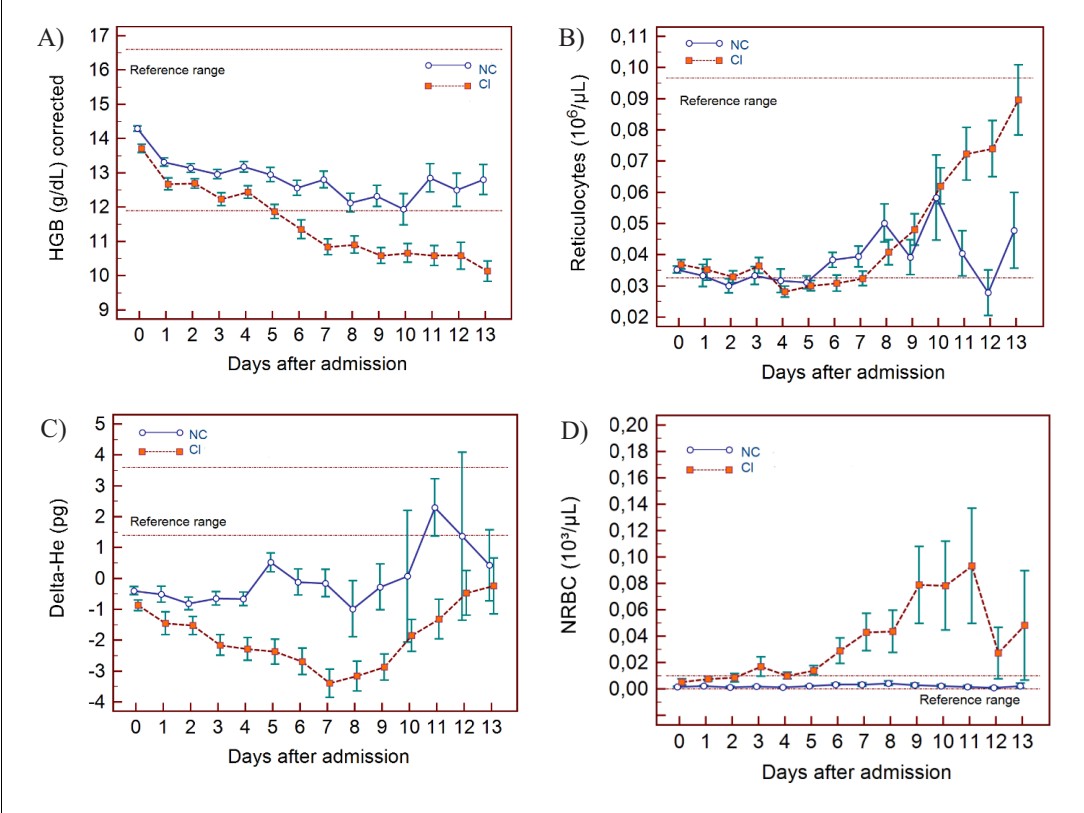

**Figure 6.** Trends of red blood cell-related parameters over 14 days of hospitalisation in critical illness (C) and non-critical (NC) patients. Note: 14 days of hospitalisation refers to day 0 (day of admission) plus the first 13 days after admission. The normal reference range is depicted by the area between the dotted horizontal lines. Vertical bars indicate standard error of the mean (SEM). (A) Haemoglobin (HGB) corrected for age and gender, (B) reticulocyte count (RET), (C) difference in haemoglobinisation of reticulocytes and red blood cells (DELTA-He), (D) nucleated red blood cells (NRBC). The number of sample measurements available per day for the trend analysis for the parameters plotted per patient group are shown in *Figure 6— source data 1*.

The online version of this article includes the following source data for figure 6:

**Source data 1.** Source data for number of measurements for each day of hospitalisation that were available per patient group for the trends of red blood cell-related parameters over 14 days of hospitalisation.

**Source data 2.** Source data for trends of corrected haemoglobin values over 14 days of hospitalisation in critical illness (CI) and non-critical (NC) patients.

**Source data 3.** Source data for trends of absolute reticulocyte count over 14 days of hospitalisation in critical illness (CI) and non-critical (NC) patients.

**Source data 4.** Source data for trends of Delta-He values over 14 days of hospitalisation in critical illness (CI) and non-critical (NC) patients.

**Source data 5.** Source data for trends of nucleated red blood cell counts over 14 days of hospitalisation in critical illness (CI) and non-critical (NC) patients.

the prognostic score ($\geq$3 or >4) over time, whereas it ranged from >4.7 to >11.6 for NLR. AUC values for platelet-to-lymphocyte ratio (PLR), LYMPH, MONO, and PLT were all low (range 0.501-0.647) with highly variable cut-off values at the different time points. The AUC comparisons for the prognostic score and these individual variables is shown in *Figure 8C*.

In investigating if the score can predict severity independent of the classical risk factors such as age and presence of comorbidities, using a Mann-Whitney test, it was found that the prognostic score was significantly higher in the CI group than the NC group across all age groups and for all age groups segregated by the presence or absence of comorbidities, with the exception of patients 84 years and older with reported comorbidities (*Table 6* and *Figure 9*). The median difference in prognostic score values between the NC and CI groups ranged from 2 to 7 points.

## Prognostic score validation

For 202 patients (*Figure 10*), 65.8% male and median age 70 years (range 22-93), with 217 (165 CI; 52 NC) CBC-DIFF-RET day 0–3 measurements available, the prognostic score gave an AUC of 0.797

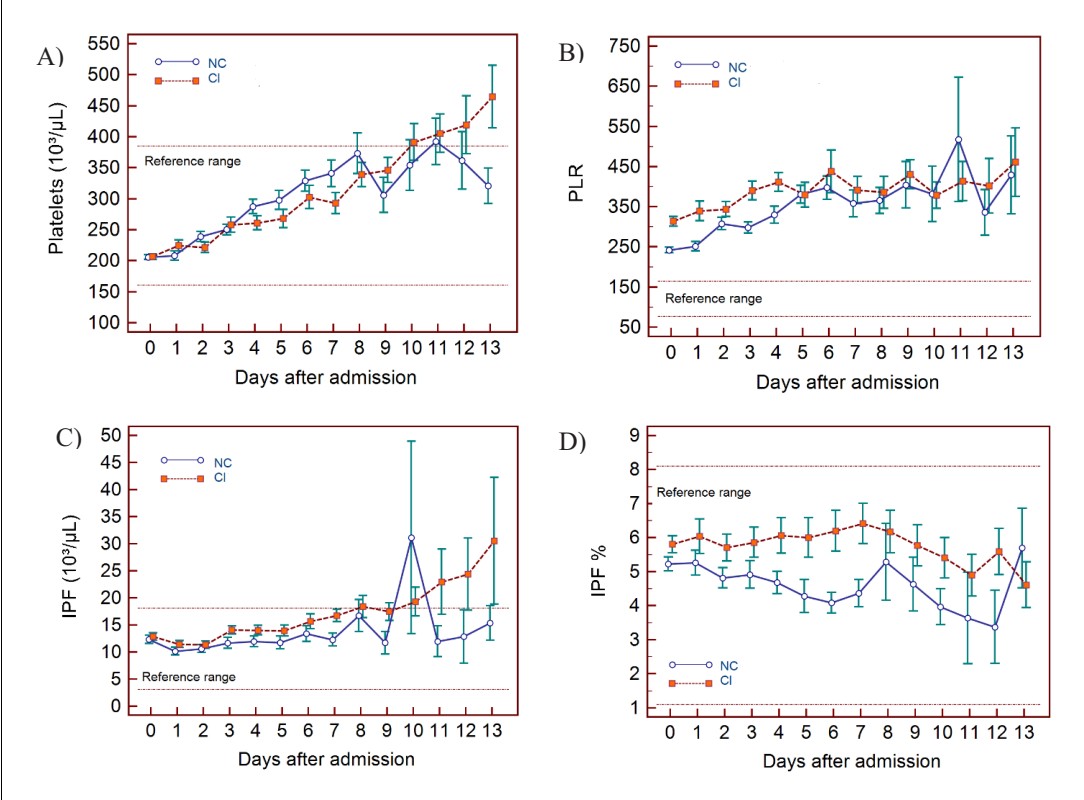

**Figure 7.** Trends of platelet parameters over 14 days of hospitalisation in critical illness (CI) and non-critical (NC) patients. Note: 14 days refers to day 0 (day of admission) plus the first 13 days after admission. The normal reference range is depicted by the area between the dotted horizontal lines. Vertical bars indicate standard error of the mean (SEM). (**A**) Platelet count (PLT), (**B**) platelet-to-lymphocyte ratio (PLR), (**C**) immature platelet count (IPF#) (**D**) immature platelet fraction (IPF%). The number of sample measurements available per day for the trend analysis for the parameters plotted per patient group are shown in *Figure 7—source data 1*.

The online version of this article includes the following source data for figure 7:

**Source data 1.** Source data for number of measurements for each day of hospitalisation that were available per patient group for the trends of platelet parameters over 14 days of hospitalisation.

**Source data 2.** Source data for trends of platelet count over 14 days of hospitalisation in critical illness (CI) and non-critical (NC) patients.

**Source data 3.** Source data for trends of platelet-to-lymphocyte count ratio over 14 days of hospitalisation in critical illness (CI) and non-critical (NC) patients.

**Source data 4.** Source data for trends of absolute immature platelet count (IPF#) over 14 days of hospitalisation in critical illness (CI) and non-critical (NC) patients.

**Source data 5.** Source data for trends of immature platelet fraction (IPF%) over 14 days of hospitalisation in critical illness (CI) and non-critical (NC) patients.

(95% CI 0.724-0.829), which is comparable to the prognostic score performance in the development cohort (AUC 0.753, 95% CI 0.723-0.781). Mortality rate was 20.8% (42/202) and outcome was correctly identified in 72% (91/127) of CI patients (development cohort 70.5%). Except for day 1, score values of the NC and CI groups over the 14-day period (718 measurements) did not overlap (SEM) (*Figure 11A*), with AUC of prognostic score superior to NLR (*Figure 11B*).

## Discussion

We showed that SARS-CoV-2 infection is accompanied by haemocytometric changes over time and that distinct haemocytometric parameters, combined in a COVID-19 prognostic score, can be used early on to identify those patients likely to deteriorate thereafter and thus may benefit from ICU admission. Moreover, our data suggest that parameters reflecting the activation or functional status

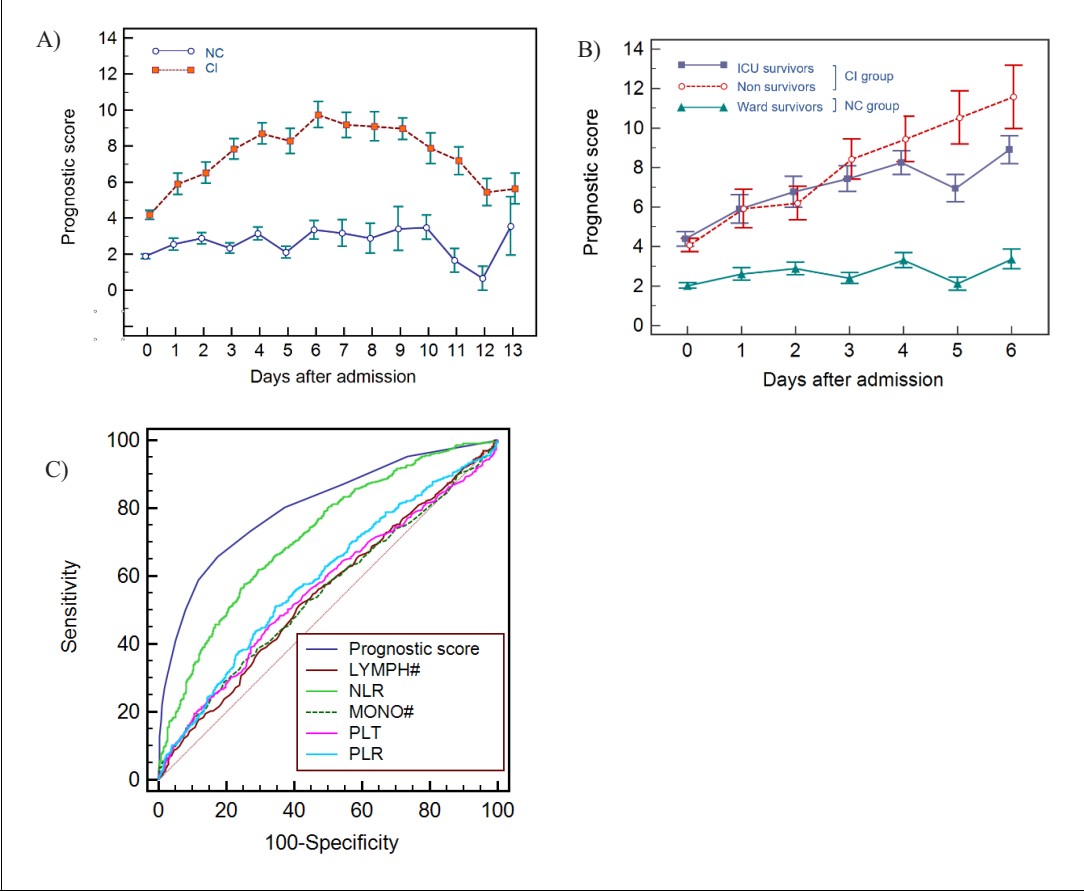

**Figure 8.** Haemocytometric COVID-19 prognostic score prediction of clinical severity in the development cohort. (**A**) Development cohort prognostic score 14-day hospitalisation time horizon (day of admission plus the first 13 days thereafter) comparing non-critical (NC) and critical illness (CI) groups. Points shown are mean values with vertical bars representing SEM (**B**) development cohort prognostic score 7-day hospitalisation time horizon comparing outcomes for the critical illness group (recovered with ICU or died) and the non-critical group (recovered without ICU). Points shown are mean values with vertical bars representing SEM, (**C**) ROC curve to compare the capability of prediction of critical illness disease progression of the prognostic score, absolute lymphocyte count (LYMPH#), neutrophil-to-lymphocyte ratio (NLR), absolute monocyte count (MONO#), platelet count (PLT) and platelet-to-lymphocyte ratio (PLR) for development cohort incorporating all measurements over the initial 14-day period of hospitalisation. The number of measurements for each day of hospitalisation that were available per patient group are shown in *Figure 8—source data 1*. There were markedly fewer measurements for the second week, notably in the NC group which may have contributed to bias.

The online version of this article includes the following source data for figure 8:

**Source data 1.** Source data for number of measurements for each day of hospitalisation that were available per patient group for prognostic score prediction of clinical severity in the development cohort.

**Source data 2.** Source data for development cohort prognostic score 14-day hospitalisation time horizon (day of admission plus the first 13 days thereafter) comparing non-critical (NC) and critical illness (CI) groups.

**Source data 3.** Source data for development cohort prognostic score 7-day hospitalisation time horizon comparing outcomes (recovered without ICU, recovered with ICU or died).

**Source data 4.** Source data for ROC curves to assess the capability of prediction of critical illness disease progression of the prognostic score, absolute lymphocyte count (LYMPH), neutrophil-to-lymphocyte ratio (NLR), absolute monocyte count (MONO), platelet count (PLT), and platelet-to-lymphocyte ratio (PLR) for development cohort incorporating all measurements over the initial 14-day period of hospitalisation.

of blood cells are better disease severity indicators than traditional parameters, such as lymphocyte or platelet counts.

In COVID-19, lymphopenia has been assigned a key role based on a higher incidence and greater suppression observed in ICU patients (*Terpos et al., 2020*). Our data indicate that lymphocyte count had no significant prognostic value (AUC 0.550; 95% CI 0.523-0.576), although relative presence of lymphocytes did, hence the incorporation of NLR, IGLR, and AS-LYMPH%/L (reflecting

**Table 3.** Haemocytometric COVID-19 prognostic score cut-off values.

| | Variable | Precondition | 1 Point | 2 Points | 3 Points | 4 Points |
|---|---|---|---|---|---|---|
| Primary Variables | IG/L*100 | IG* $\geq$ 0,09 | $\geq$10 | $\geq$20 | $\geq$40 | $\geq$45 |
| | NLR | none | [<7,7 and LYMPH† < 1,07 and N/L² $\geq$7,5] or [$\geq$7,7 < 16,5 and LYMPH† < 0,65] | [$\geq$7,7 < 16,5 and LYMPH† $\geq$ 0,65] | [$\geq$7,7 < 16,5 and LYMPH† $\geq$ 1,07] | $\geq$16,5 |
| | RE-MONO/M [%] | none | [$\geq$5 < 15 and RE-MONO‡ < 0,03] | [$\geq$5 < 15 and Re-MONO‡ $\geq$ 0,03] | n/a | $\geq$15 |
| | AS-LYMPH/L [%] | none | [$\geq$5 and LYMPH§ < 10] | $\geq$10 | n/a | $\geq$15 |
| | DELTA-He [pg] | RET¶ $\geq$ 6.0 | [< −1 $\geq$ −2] or [<0,4 $\geq$ −1 and RET¶ $\geq$ 20.0] | [< −2 $\geq$ −4] | n/a | < −4 |
| | NRBC [/µL] | HGB** $\geq$ 9 and RET¶ < 90 | n/a | $\geq$ 20 | n/a | $\geq$40 |
| Secondary Variables | HGB [g/dl] | none | $\geq$17 | | | |
| | HYPO-He [%] | Micro-R†† < 10 | $\geq$1,9 | | | |
| | PLT [10³/µL] | none | <85 | | | |
| | IPF# [10³/µL] | none | $\geq$25 | | | |

Note: For the primary variables, one point = value above the cut-off value for the best AUC; two points = value above the cut-off value for the best AUC and $\geq$80% specificity; three points = value above the cut-off value for the best AUC and >90% specificity; and four points = value above the cut-off value for the best AUC and >95% specificity. The cut-off values for the secondary variables were chosen exclusively based on observed extremes of values in critical disease, with the maximum award of 1 point per variable.

The prognostic score values were calculated automatically using a pre-set algorithm, using the above cut-off values to assign points per individual parameter. The aim is to have the formula for the calculation incorporated into the Laboratory Information System software in use in individual laboratories.

Abbreviations: IGLR, immature granulocytes-to-lymphocyte ratio; NLR, neutrophil-to-lymphocyte ratio; AS-LYMPH%/L, antibody synthesising lymphocytes as a proportion of lymphocytes; RE-MONO%/M, reactive monocytes as a proportion of monocytes; DELTA-He, difference in haemoglobinisation of reticulocytes and red blood cells; NRBC, absolute nucleated red blood cell count; HGB, haemoglobin; HYPO-He, percentage of red blood cells that are hypochromic; PLT, platelet count; IPF, absolute immature platelet count; N/L², neutrophil-to-lymphocyte squared ratio.

* Unit of measure for IG (absolute immature granulocyte count) is x 10³/µL.

† Unit of measure for LYMPH (absolute lymphocyte count) is x 10³/µL.

‡ Unit of measure for RE-MONO (absolute reactive monocyte count) is x 10³/µL.

§ Unit of measure for LYMPH% (percentage lymphocyte count).

¶ Unit of measure for RET (absolute reticulocyte count) is x 10³/µL.

** Unit of measure for HGB (haemoglobin) is g/dL.

†† Unit of measure for Micro-R is %.

lymphoplasmacytoid lymphocytes [*Linssen et al., 2007*]) in the prognostic score. Increased lymphocyte activation has been documented in COVID-19 (*Fan et al., 2020*; *Chong et al., 2020*). Our findings concur with a previous study (*Yip et al., 2020*) that reported AS-LYMPH as a strong predictor of clinical severity in COVID-19 patients. Furthermore, a specific "hourglass" appearance on the WBC scattergram on Sysmex analysers, representative of lymphoplasmacytoid lymphocytes, was reported to have a high positive predictive value to detect COVID-19 (*Osman et al., 2020*).

In contrast to lymphocytes, neutrophils, including precursors, have a tendency to increase in COVID-19 (*Mitra et al., 2020*). IGs, representing metamyelocytes, myelocytes,, and promyelocytes (*Briggs et al., 2011*), were commonly present in our patient population, especially in those more severely ill. The importance of IGs in management of sepsis has been reported (*Ayres et al., 2019*; *Nierhaus et al., 2013*). Of note, increases in neutrophil counts and neutrophil activity (NEUT-RI) were dissimilar, unlike observations in bacterial infections (*Park et al., 2015*). RE-MONO%/M is abnormal in critical illness cases only, in line with a previous report attributing a key role for monocytes and macrophages in severe COVID-19 (*Merad and Martin, 2020*).

**Table 4.** Development cohort prognostic score values by day based on clinical severity group, initial management, and outcome.

| day | Not hospitalised | | Hospitalised with admission to a general ward on day of initial presentation (non-ICU) | | | | Hospitalised with admission directly to ICU on day of initial presentation | | | |
| --- | --- | --- | --- | --- | --- | --- | --- | --- | --- | --- |
| | Mild Median score* | n | Recovered Median score* | n | Died Median score* | n | Recovered Median score* | n | Died Median score* | n |
| 0 | 0.5 (0.0–1.0) | 34 | 1.0 (1.0–2.0) | 243 | 3.0 (2.0–4.0) | 86 | 4.0 (3.0–6.0) | 70 | 4.0 (2.6–8.4) | 21 |
| 1 | | | 2.0 (1.5–3.0) | 59 | 4.0 (1,5–6.7) | 15 | 6.5 (3.0–9.3) | 24 | 7.0 (1.5–14,1) | 10 |
| 2 | | | 2.0 (1.0–3.0) | 96 | 5.0 (4.0–6.0) | 23 | 7.0 (6.0–8.9) | 37 | 12.5 (1.8–16.6) | 8 |
| 3 | | | 2.0 (1.0–3.0) | 75 | 5.0 (4.4–7.6) | 16 | 8.0 (7.0–9.0) | 31 | 12.5 (7.8–18.3) | 8 |
| 4 | | | 3.0 (2.0–4.0) | 59 | 7.0 (6.0–8.7) | 15 | 8.0 (6.0–10.8) | 30 | 15.0 (3.8–20.4) | 8 |
| 5 | | | 2.0 (1.0–3.0) | 37 | 12.0 (6.5–14.9) | 13 | 6.5 (4.8–8.3) | 34 | 7.0 (0.8–19.1) | 8 |
| 6 | | | 3.0 (2.0–4.0) | 37 | 11.0 (6.1–14.8) | 9 | 9.0 (7.0–11.0) | 29 | 14.0 (na†) | 5 |
| 7 | | | 2.0 (0.3–6.7) | 23 | 11.0 (5.2–15.4) | 6 | 8.0 (6.0–11.2) | 29 | 8.0 (5.8–12.4) | 11 |

Note: Recovered refers to patients that survived and were discharged from hospital. By definition, 'Mild' patients are not hospitalised and therefore only day 0 prognostic score values are available.

* Median score refers to the median value obtained for the haemocytometric COVID-19 prognostic score for the patient group for patients with the same length of hospitalisation on the day of measurement. Values in brackets represent the 95% CI for the median.

† Sample size was too small to calculate 95% CI.

The online version of this article includes the following source data for Table 4:

Source data 1. Source data for development cohort prognostic score values by day based on clinical severity group, initial management, and outcome.

Erythropoietic changes have been reported, mostly showing low HGB levels (*Fan et al., 2020*; *Lippi and Plebani, 2020b*; *Sun et al., 2020c*). We found that whilst HGB levels decrease in COVID-19 patients, the erythropoietic response to anaemia, indicated by RET and reticulocyte production index (data not shown), were mostly normal. Haemoglobinisation of reticulocytes, as indicated by negative DELTA-He levels, is however significantly compromised, specifically in more severe cases, possibly due to ongoing inflammation (*Weimann et al., 2016*). NRBCs are absent in peripheral blood of healthy adults. Their presence, without reticulocytosis, in severe COVID-19 cases indicates haematopoietic stress, probably due to prolonged hypoxia or inflammation (*Danise et al., 2011*). Furthermore, NRBCs, were reported as a marker of disease severity in ARDS patients, indicating a higher risk of death (*Menk et al., 2018*).

Contrary to other studies (*Henry et al., 2020*; *Jiang et al., 2020*; *Wang et al., 2020b*; *Yang et al., 2020b*), PLT at presentation were similar between CI and NC cases, mostly within normal limits with no sign of increased platelet consumption as IPF also remained normal. PLT, and IPF#, tended to increase with disease severity. Higher PLT have been previously reported in severe COVID-19 (*Qu et al., 2020*). So thrombocytosis, more than thrombocytopenia may be linked to severe COVID-19 which is in contrast to guidelines to identify severe pneumonia (*Metlay et al., 2019*).

A recent meta-analysis (*Zeng et al., 2020*) concluded that severe COVID-19 patients had higher neutrophil counts and NLR, and lower lymphocyte counts than those with non-severe COVID-19, and that these basic parameters might help clinicians to predict the severity and prognosis of COVID-19. Although our findings concur with their observed clinical severity-based WBC differences, the discriminating power, early on during hospitalisation and thus value to determine prognosis, was insufficient. A previous report about the prognostic value of NLR (*Liu et al., 2020b*) is also not supported by our data. Altogether, our findings indicate that new parameters, reflecting functional status of blood cells, are more frequently outside reference ranges in COVID-19 than classical parameters such as lymphocytes, neutrophils or platelets. However, none of the measured parameters, traditional or novel, alone could discriminate patients based on disease severity. The prognostic score we developed used multiple parameters, representing the three haemopoietic cell lines, with the aim to distinguish CI from NC COVID-19 patients. Different cell lines may not all be equally affected by COVID-19 at the same time. Thus, it is not unexpected that multiple parameters

**Table 5.** Receiver Operator Characteristics (ROC) curve comparisons between the haemocytometric COVID-19 prognostic score versus other parameters.

| | Prognostic score | LYMPH (x10³/μL) | NLR | MONO (x10³/μL) | PLT (x10³/μL) | PLR |
|---|---|---|---|---|---|---|
| Day* < 4 [n† = 859] | | | | | | |
| [AUC] | 0.753 | 0.591 | 0.709 | 0.516 | 0.537 | 0.602 |
| [95% CI] | 0.723 to 0.781 | 0.558 to 0.624 | 0.678 to 0.740 | 0.482 to 0.550 | 0.503 to 0.570 | 0.568 to 0.635 |
| [cut-off value] | >3 | <0.810 | >7.7 | <0.400 | >234 | >308 |
| [P value vs prognostic score] | na | <0.0001 | 0.0066 | <0.0001 | <0.0001 | <0.0001 |
| Day < 14 [n = 1423] | | | | | | |
| [AUC] | 0.806 | 0.55 | 0.718 | 0.551 | 0.563 | 0.591 |
| [95% CI] | 0.784 to 0.826 | 0.523 to 0.576 | 0.694 to 0.741 | 0.525 to 0.577 | 0.537 to 0.589 | 0.565 to 0.617 |
| [cut-off value] | >4 | <0.810 | >7.1 | >0.640 | >268 | >308 |
| [P value vs prognostic score] | na | <0.0001 | <0.0001 | <0.0001 | <0.0001 | <0.0001 |
| Day 0 [n = 454] | | | | | | |
| [AUC] | 0.722 | 0.603 | 0.683 | 0.506 | 0.537 | 0.605 |
| [95% CI] | 0.678 to 0.763 | 0.556 to 0.648 | 0.638 to 0.726 | 0.459 to 0.553 | 0.490 to 0.584 | 0.558 to 0.650 |
| [cut-off value] | >3 | <0.780 | >7.6 | >0.700 | >234 | >375 |
| [P value vs prognostic score] | na | 0.0001 | 0.0660 | <0.0001 | <0.0001 | 0.0001 |
| Day 1 [n = 109] | | | | | | |
| [AUC] | 0.737 | 0.557 | 0.664 | 0.511 | 0.555 | 0.602 |
| [95% CI] | 0.644 to 0.817 | 0.459 to 0.652 | 0.567 to 0.751 | 0.413 to 0.608 | 0.457 to 0.651 | 0.504 to 0.695 |
| [cut-off value] | >4 | <0.620 | >11.6 | >0.180 | >218 | >358 |
| [p value vs. prognostic score] | na | 0.0036 | 0.1132 | 0.0012 | 0.0045 | 0.0265 |
| Day 2 [n = 164] | | | | | | |
| [AUC] | 0.739 | 0.572 | 0.714 | 0.575 | 0.515 | 0.526 |
| [95% CI] | 0.665 to 0.804 | 0.492 to 0.649 | 0.638 to 0.782 | 0.495 to 0.652 | 0.436 to 0.594 | 0.446 to 0.604 |
| [cut-off value] | >4 | <1.020 | >5.1 | <0.470 | <118 | >217 |
| [P value vs prognostic score] | na | 0.0006 | 0.4780 | 0.0026 | <0.0001 | 0.0001 |
| Day 3 [n = 132] | | | | | | |
| [AUC] | 0.875 | 0.602 | 0.827 | 0.501 | 0.582 | 0.647 |
| [95% CI] | 0.806 to 0.926 | 0.514 to 0.686 | 0.751 to 0.887 | 0.413 to 0.590 | 0.493 to 0.667 | 0.559 to 0.728 |
| [cut-off value] | >3 | <0.810 | >7,3 | <0.680 | >255 | >280 |
| [p value vs prognostic score] | na | <0.0001 | 0.1860 | <0.0001 | <0.0001 | <0.0001 |
| Day 4 [n = 112] | | | | | | |
| [AUC] | 0.867 | 0.604 | 0.778 | 0.572 | 0.531 | 0.583 |
| [95% CI] | 0.790 to 0.924 | 0.507 to 0.695 | 0.690 to 0.851 | 0.475 to 0.665 | 0.434 to 0.626 | 0.486 to 0.676 |
| [cut-off value] | >4 | <0.710 | >7.7 | <0.270 | <238 | >330 |
| [p value vs prognostic score] | na | <0.0001 | 0.0070 | <0.0001 | <0.0001 | <0.0001 |
| Day 5 [n = 92] | | | | | | |
| [AUC] | 0.877 | 0.578 | 0.753 | 0.578 | 0.599 | 0.506 |
| [95% CI] | 0.792 to 0.936 | 0.471 to 0.680 | 0.652 to 0.837 | 0.470 to 0.680 | 0.492 to 0.700 | 0.400 to 0.612 |
| [cut-off value] | >4 | <1.140 | >6.5 | <0.580 | <227 | <350 |
| [p value vs prognostic score] | na | <0.0001 | 0.0030 | <0.0001 | <0.0001 | <0.0001 |
| Day 6 [n = 80] | | | | | | |
| [AUC] | 0.875 | 0.517 | 0.688 | 0.511 | 0.589 | 0.545 |

*Table 5 continued on next page*

Table 5 continued

| | Prognostic score | LYMPH (x10³/µL) | NLR | MONO (x10³/µL) | PLT (x10³/µL) | PLR |
|---|---|---|---|---|---|---|
| [95% CI] | 0.782 to 0.938 | 0.403 to 0.630 | 0.575 to 0.787 | 0.397 to 0.625 | 0.474 to 0.698 | 0.430 to 0.657 |
| [cut-off value] | >4 | <1.620 | >7.0 | >0.640 | <277 | <153 |
| [p value vs prognostic score] | na | <0.0001 | <0.0001 | <0.0001 | <0.0001 | <0.0001 |
| Day 7 [n = 100] | | | | | | |
| [AUC] | 0.856 | 0.633 | 0.724 | 0.561 | 0.596 | 0.549 |
| [95% CI] | 0.772 to 0.918 | 0.531 to 0.727 | 0.625 to 0.808 | 0.458 to 0.660 | 0.493 to 0.693 | 0.446 to 0.649 |
| [cut-off value] | >3 | <0.940 | >4.7 | <0.310 | <298 | >311 |
| [p value vs prognostic score] | na | 0.0002 | 0.0044 | <0.0001 | <0.0001 | <0.0001 |

Note:"Prognostic score' refers to the haemocytometric COVID-19 prognostic score comprised of NLR, IGLR, RE-MONO%/M, AS-LYMPH%/L, DELTA-He, NRBC, HGB, Hypo-He, PLT, and IPF#.

Abbreviations: NLR, neutrophil-to-lymphocyte ratio; IGLR, immature granulocyte-to-lymphocyte ratio; RE-MONO%/M, reactive monocytes as percentage of total monocyte count; AS-LYMPH%/L, antibody-synthesising lymphocytes as percentage of total lymphocyte count; DELTA-He, difference in haemoglobinisation of reticulocytes and red blood cells; NRBC, absolute nucleated red blood cell count; HGB, haemoglobin; Hypo-He, percentage of red blood cells that are hypochromic; PLT, platelet count; IPF#, absolute immature platelet count; LYMPH, absolute lymphocyte count; MONO, absolute monocyte count; PLR, platelet-to-lymphocyte ratio; AUC, area under the curve; CI, confidence interval; na, not applicable.

* 'Day' refers to the number of days of hospitalisation of patient at time of blood sample measurement. Day 0 refers to the day of first presentation at hospital.

† 'n' refers to the number of complete profile (complete blood and differential count and reticulocyte channel) sample measurements available at a particular time point and included in the ROC curve analysis.

The online version of this article includes the following source data for Table 5:

Source data 1. Source data for Receiver Operator Characteristics (ROC) curve comparisons between the haemocytometric COVID-19 prognostic score versus other parameters.

together are better at predicting disease severity, as duration of symptoms at presentation differs widely in individual patients.

The patient group is inherently heterogeneous as health-seeking behaviour may vary widely between patients: some patients may present early whilst others present late at health facilities. Furthermore, recall when symptoms started may be unclear at time of presentation to the hospital. Therefore, a patient on day 3 of hospitalisation could be in an earlier phase of infection than a patient on day 0. In this regard, as the intended use of the prognostic score was to identify at an early stage (once medical care had been sought) who is likely to deteriorate, to keep it as simple and practical as possible, we analysed the data from day 0–3 as a collective dataset for the purpose of identifying which parameters, and at what cut-off values, should be incorporated in the score. Likewise, we did not compare score values from the early phase of hospitalisation (days 0–3) with the later phase (days 4–13) in keeping with the intention to predict outcome early on before signs of critical illness may become clinically overt. We did however plot the data over time to document the patterns as this may be useful in the future to assess response to specific COVID-19 therapies.

The prognostic score correctly identified 70.5% of these patients during first 3 days after hospital admission, with similar performance confirmed in the validation cohort (72.0%). Prognostic score trends over 14 days confirmed a stable clinical course in NC patients and disease progression in CI patients, peaking on day 6 (sensitivity 93%). In the development cohort there was a distinct progressive upward trend of prognostic score values from day 0, peaking on day 6 in the CI group. In the validation cohort, the score also peaked on day 6, but the day 0 score value started relatively high, dropping on day 1, giving the appearance of convergence with the NC group on day 1. The validation cohort was comprised of patients from only two hospitals. Study enrolment at these two hospitals took place at different times of the pandemic, during which time ICU capacity was ramped up significantly. In this regard we speculate that ICU access differed at the two hospitals, hence patients with a similar degree of disease severity may have been admitted to ICU at one hospital and remained on the general ward at the other. Furthermore, all day 0 patients in the CI group were from a single hospital, and numbers were small relative to the development cohort. Our ethics approval did not permit us to review the individual patient clinical records of six patients in the CI

**Table 6.** Mann-Whitney test for significance of the difference in prognostic score between critical illness (CI) and non-critical (NC) patients.

| Age range (years) | 25-34[†] | 35–44 | 45–54 | 55–64 | 65–74 | 75–84 | >84 |
|---|---|---|---|---|---|---|---|
| CI vs NC | p=0.0030 | p<0.0001 md = 7 | p<0.0001 md = 6 | p<0.0001 md = 4 | p<0.0001 md = 5 | p<0.0001 md = 3 | p=0.0006 md = 2 |
| CI vs NC with reported comorbidities[*] | Sample size insufficient | p=0.0046 md = 5 | p=0.0094 md = 3 | p<0.0001 md = 5 | p<0.0001 md = 5 | p<0.0001 md = 4 | p=0.0748 md=2[‡] |
| CI vs NC with comorbidities[*] reported as 'none' | Sample size insufficient | p=0.0011 md = 7 | p<0.0001 md = 5 | p<0.0001 md = 4 | p=0.0305 md = 2 | p<0.0001 md = 5 | p=0.0339 md = 3 |

Note: The data is shown per age group for all patients as well as for patients segregated according to the presence or absence of reported comorbidities.

'md' refers to the Hodges-Lehmann median difference in prognostic score values between the NC and CI groups in each age category.

* Patients that were enrolled into the study from the two hospitals that did not report on comorbidities were excluded from the analysis.

† Sample size was too small for analysis in this age group segregated by comorbidities.

‡ Patients older than 84 years and with reported comorbidities were the only group where the difference between CI and NC groups was not significant.

The online version of this article includes the following source data for Table 6:

Source data 1. Mann–Whitney test for significance of the difference in prognostic score between critical illness (CI) and non-critical (NC) patients.

group that already had very high (10–16 points) prognostic score values on day 0, four of whom also had no further follow-up samples which would have brought the score value averages on the days following admission, upwards. Any such inter-hospital variability would have been masked in the development cohort, because of the much larger sample size and representation from nine hospitals, compared to the validation cohort where such differences may have been more readily exposed. We therefore speculate that the high average day 0 score value patients in the validation cohort are outliers, and that the day 1 data is more representative of the expected prognostic score values in the validation cohort data set.

As has been widely reported in the literature, our data too shows that males were predominantly affected, and that disease severity was associated with increasing age and presence of comorbidities in general (*Table 1*, *Figure 2*). However, not all young patients had a mild course, and not all old patients with comorbidities were critical. Systemic inflammation is an important factor driving disease severity. Our prognostic score, incorporating the activation status of immune cells, may therefore have additional value, especially on an individual patient level, over classical risk factors such as

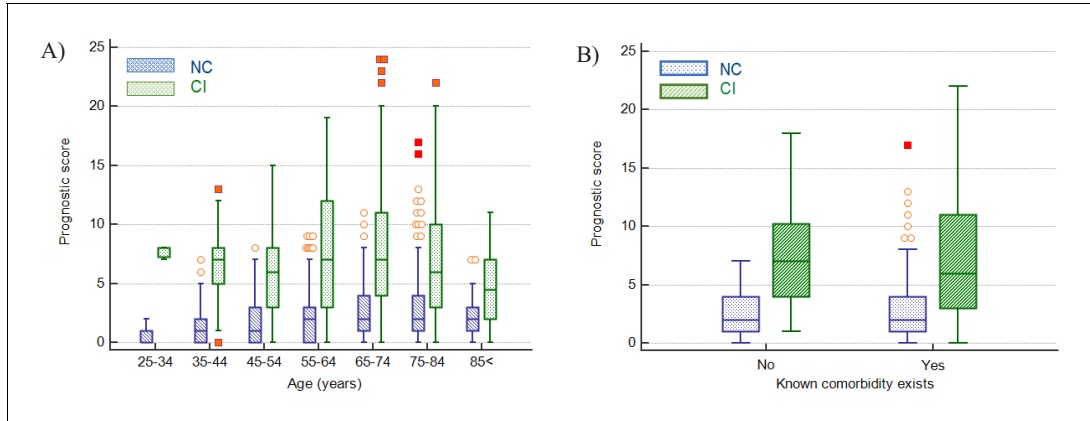

**Figure 9.** Impact of age and presence of comorbidities on prediction of disease severity. (A) Box and whisker plots of prognostic score values for NC and CI groups segregated by age. The prognostic score can predict severity independent of age, therefore potentially assisting in identifying young patients at risk for severe disease progression as well as older patients not at risk. (B) Box and whisker plots of prognostic score values for NC and CI groups segregated by comorbidities in the 75-84-year-old group, as an illustrative example. The prognostic score is significantly higher in patients with severe disease progression independent of the presence of comorbidities. Please refer to *Table 6* for more detailed information on all age groups.

The online version of this article includes the following source data for figure 9:

Source data 1. Impact of age on prediction of disease severity.

**Table 7.** Sysmex XN-1000 haematology analyser parameters used in this study.

| Parameter name | Standard parameters* | Parameter description | Measurement profile |
|---|---|---|---|
| AS-LYMPH | | Antibody synthesising lymphocyte count (this is a subset of RE-LYMPH) | CBC-DIFF |
| AS-LYMPH % | | Antibody-synthesising lymphocyte percentage of total white blood cell count | CBC-DIFF |
| AS-LYMPH %/L | | Antibody-synthesising lymphocytes as a percentage of lymphocytes | CBC-DIFF |
| BASO | Yes | Basophil count | CBC-DIFF |
| Delta-He | | Standard parameter calculated by the equation RET-He minus RBC-He | RET |
| EO | Yes | Eosinophil count | CBC-DIFF |
| HCT | Yes | Haematocrit | CBC-DIFF |
| HGB | Yes | Haemoglobin concentration | CBC-DIFF |
| HYPO-He | | The ratio of the count in the low-level area of the forward scattered light signal in the RBC area of the RET scattergram, to mature red blood cells (% of hypochromic red blood cells of total red blood cells) | RET |
| IG | | Immature granulocyte count | CBC-DIFF |
| IPF | | Immature platelet fraction (% of immature platelets of total platelet count) | RET (PLT-F)† |
| IPF# | | Immature platelet fraction count (immature platelet absolute count) | RET (PLT-F)† |
| LYMPH | Yes | Lymphocyte count | CBC-DIFF |
| MCH | Yes | Mean corpuscular haemoglobin | CBC-DIFF |
| MCHC | Yes | Mean corpuscular haemoglobin concentration | CBC-DIFF |
| MCV | Yes | Mean corpuscular volume | CBC-DIFF |
| MicroR | | Micro RBC ratio (proportion of small red blood cells (RBCs) as a % of total RBCs) | RET |
| NEUT | Yes | Neutrophil count | CBC-DIFF |
| MONO | Yes | Monocyte count | CBC-DIFF |
| NEUT-GI | | Neutrophil granularity index (reactivity of neutrophils (cytoplasmic granulation)) | CBC-DIFF |
| NEUT-RI | | Neutrophil reactivity index (reactivity of neutrophils (metabolic activity)) | CBC-DIFF |
| NRBC | | Nucleated red blood cell count | CBC-DIFF |
| PLT | Yes | Platelet count | CBC-DIFF |
| RBC | Yes | Red blood cell (erythrocyte) count | CBC-DIFF |
| RBC-He | | Mature RBC haemoglobin equivalent (optical measurement of red blood cell haemoglobinisation) | RET |
| RE-LYMPH | | Reactive lymphocyte count | CBC-DIFF |
| RE-LYMPH % | | Reactive lymphocyte percentage of total white blood cell count | CBC-DIFF |
| RE-LYMPH %/L | | Reactive lymphocytes as a percentage of lymphocytes | CBC-DIFF |
| RE-MONO | | Number of monocytes with a side fluorescent signal > 150 channels representing activated monocytes | CBC-DIFF |
| RE-MONO %/M | | Reactive monocytes as a percentage of monocytes | CBC-DIFF |
| RET# | Yes | Reticulocyte count | RET |
| RET-He | | Reticulocyte haemoglobin equivalent (optical measurement of reticulocyte haemoglobinisation) | RET |
| WBC | Yes | White blood cell (leukocyte) count | CBC-DIFF |

Note: The availability of individual parameters as either diagnostic (IVD) or research use only (RUO) is dependent on regulatory approval status which differs across regions.

* Standard parameters available in the complete blood count (CBC), differential (DIFF) or reticulocyte measurement (RET) channels across multiple manufacturer haematology analyser platforms.

† Depending on region, the immature platelet count (IPF# and IPF%) are obtained either from the RET or PLT-F channel. PLT-F refers to fluorescent optical platelet measurement.

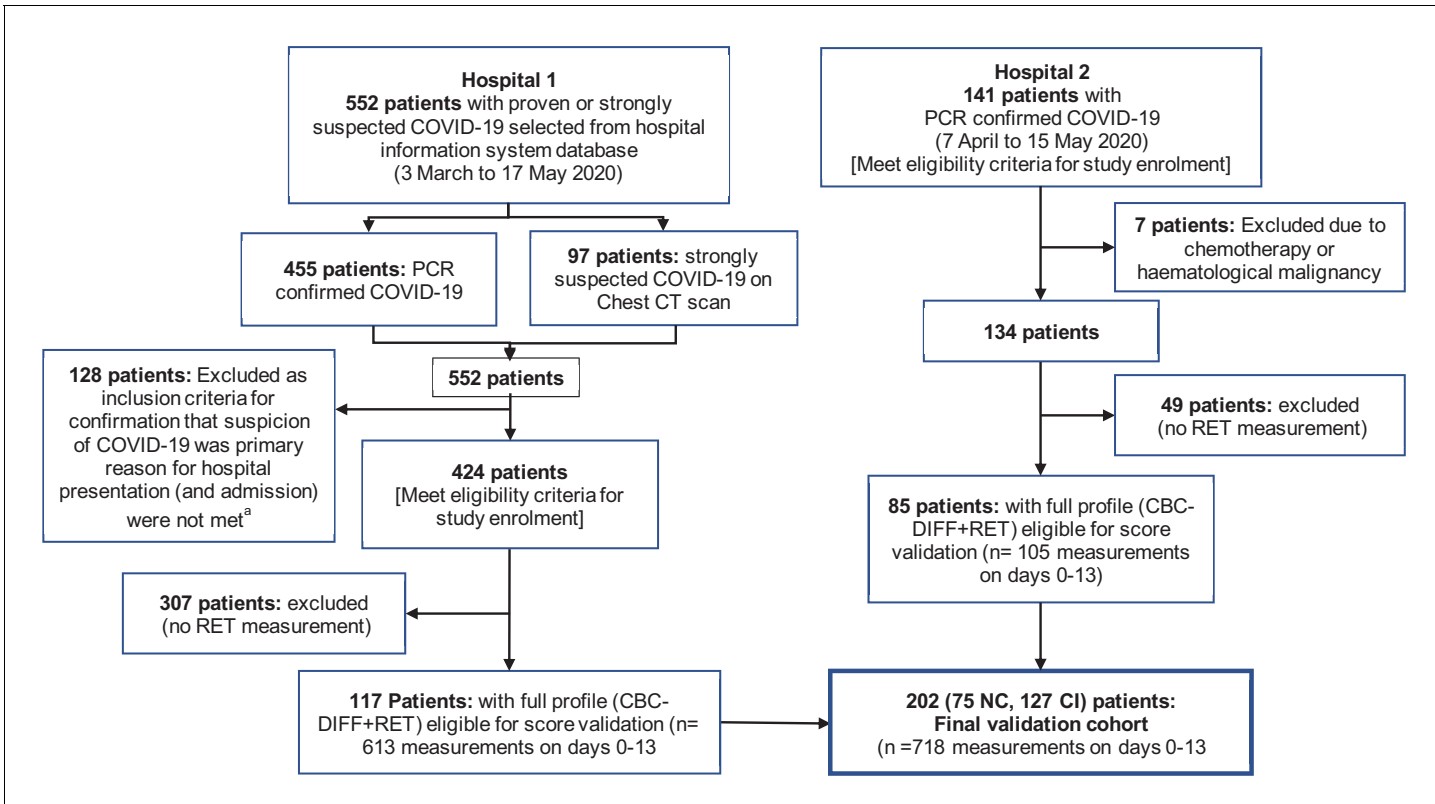

**Figure 10.** Flow chart showing inclusion and exclusion of validation cohort patient. Abbreviations: NC, non-critical patient group; CI, critical illness patient group. (a) The following criteria were used to ensure selection of only those patients for whom the primary presentation at hospital was related to COVID-19: Emergency department location on day 0, provisional diagnosis of pneumonia, if admitted, with admission to a general ward, internal medicine, ICU or anaesthesia (critical care).

age, gender and comorbidities in discriminating between NC and CI patients, (*Table 6*, *Figure 9*). As such, our prognostic score may assist in identifying any patients at risk for severe disease progression, being young or old, male or female, with or without comorbidity and by doing so, support individualised treatment decisions with objective data.

Notably, the mortality rate was relatively high in patients on the general ward, comparable to that of patients admitted to ICU, possibly due to ICU bed shortages or unfamiliarity with COVID-19 at that time. Whatever reason, we assumed that the need for more intensive treatment should have been considered for all patients that died in the general ward. Once a patient is overtly critically ill, clinical judgement will suffice to prioritise intensive care for such a patient. Importantly, our score strives to identify the deranged immune response as a harbinger of criticality (organ failure) before such organ failure is clinically evident. Our data (*Figure 8B*) shows that prognostic score values of patients with a critical clinical course (CI group) were indistinguishable based on outcome, namely whether they recovered or died, confirming that the clinical applicability of our score is to predict on an individual patient level who is likely to have a critical clinical course, but not to predict mortality.

In our study, the prognostic score was calculated retrospectively using the haemocytometric data from each individual sample measurement exported into Microsoft Excel (Microsoft Corp., USA). For future clinical practice, we envisage that the prognostic score will be automatically calculated by the laboratory information system and that the result is given on request, together with complete blood count analysis. Besides serving as a risk stratification tool for clinical decision making early on, we postulate that the prognostic score, which provides a snapshot in time of the phase of an individual's immune response, may be promising as an aid to patient selection for future clinical trials exploring therapeutic options for COVID-19.

Most previous risk score development studies were limited by small sample size, single centre, univariate analysis, and lack of a validation cohort. In contrast, the 4C mortality score, developed

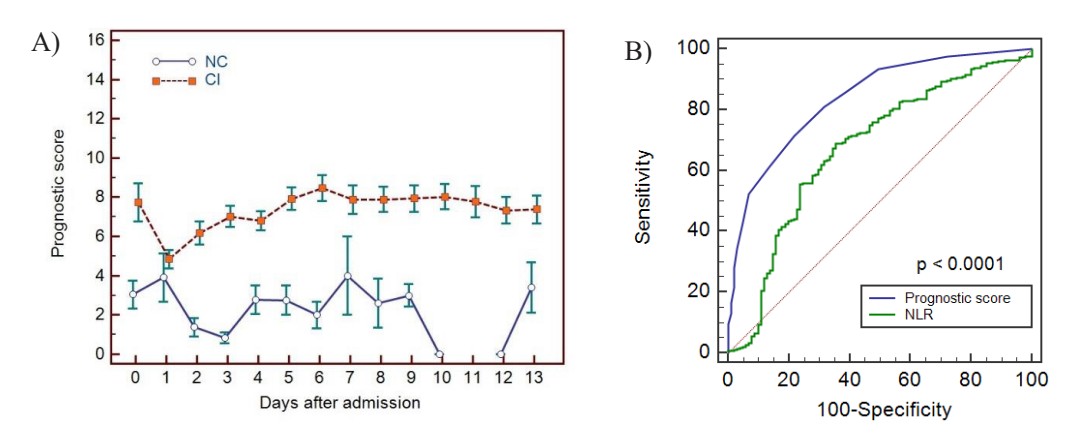

**Figure 11.** Haemocytometric COVID-19 prognostic score prediction of clinical severity in the validation cohort. (**A**) Validation cohort prognostic score 14-day hospitalisation time horizon comparing non-critical (NC) and critical illness (CI) groups, (**B**) ROC curve comparisons of prognostic score and NLR over 14 days. The prognostic score AUC (0.838, 95%CI 0.809-0.864) was significantly higher (P<0.0001) than the NLR AUC (0.673,95% CI 0.637-0.707). The number of measurements for each day of hospitalisation that were available per patient group are shown in *Figure 11—source data 1*. Overall, there were relatively few measurements per day for the NC group which has contributed to greater variance per time point.

The online version of this article includes the following source data for figure 11:

**Source data 1.** Source data for number of measurements for each day of hospitalisation that were available per patient group for prognostic score prediction of clinical severity in the validation cohort.

**Source data 2.** Source data for haemocytometric COVID-19 prognostic score prediction of clinical severity in the validation cohort over 14 day hospitalisation time horizon comparing non-critical (NC) and critical illness (CI) groups.

**Source data 3.** Source data for ROC curve comparisons of prognostic score and NLR over 14 days.

and validated by the International Severe Acute Respiratory and Emerging Infections Consortium (ISARIC), incorporating patient demographic information (age, sex, comorbidities), clinical observations (respiratory rate, peripheral oxygen saturation, level of consciousness) and blood parameters (C-reactive protein (CRP), urea level), incorporated more than 35,000 and 22,000 patients into the development and validation cohorts respectively, and showed better performance at predicting mortality in COVID-19 patients than all previously published scores (*Knight et al., 2020*). The ISARIC investigators conclude that their score is easy to use as commonly available at hospital admission. Our score has several advantages: firstly, only objective measurements of a haematology analyser are used whilst the ISARIC 4C mortality score uses clinical observation parameters which may be subject to interpretation; secondly, our score is aimed to be automatically generated whilst 4C mortality score needs to be calculated; thirdly, different laboratory measurements (oxygen saturation, CRP, urea) are required to calculate the 4C mortality score, whilst our score uses the globally most commonly requested laboratory examination for patients attending health facilities; fourthly, haematology analysers are widely available globally more so than CRP measurements; and fifthly, the 4C mortality score is aimed at patients admitted to the hospital, whilst our score was developed for patients presenting at the hospital (of whom some were never admitted).

The strength of our study is the inclusion of a relatively large group of confirmed COVID-19 cases from multiple centres and countries, including a validation cohort, Furthermore, we believe that an advantage of our prognostic score is that all input data required to calculate the score value are generated from a single haematology profile test, which is the most common routinely requested baseline blood test in all patients globally.

A limitation of our study is its retrospective nature, as data retrieved from hospital records were sometimes incomplete. Our study was performed at a time that COVID-19 was a new disease entity that constrained the health care in many of the participating centres which may have affected management decisions and therefore study outcome parameters. Data from out-patient settings including more mild cases, and from nursing homes that usually accommodate high risk patients, are needed. Furthermore, clinical data collection was limited, including comorbidity affecting COVID-19

susceptibility and ICU admission decisions, notably as the demand for ICU beds was greater than the availability at the time of our study. Importantly, the conditions of our fast-tracked ethics clearance to facilitate rapid study initiation did not permit data collection about bacterial superinfections and medication (antibiotics, corticosteroids), whilst these factors may affect outcome and haemocytometric parameters.

Finally, our prognostic score includes Sysmex unique parameters. This is a limitation as the score is not universally applicable to all haematology analysers, although the concept is transferable (see *Supplementary file 2* for parameters available on other manufacturer haematology platforms). However, it is the very ability to quantify blood cell activation, a reflection of the general immune response status of an individual, that has rendered our prognostic score (which incorporates cell activation parameters such as reactive monocytes and antibody-synthesising lymphocytes amongst others) better than using only standard parameters, such as neutrophil-to-lymphocyte ratio, which are universally available on all systems, at least in our patient dataset.

## Conclusions

Our finding of potential usefulness of extended haemocytometry may be impactful as Sysmex haematology analysers are widely available. Haematology blood profile requests are common, inexpensive, quick, highly standardised, quality-controlled baseline tests. Furthermore, this investigation is requested in febrile patients and those with non-febrile conditions. As the latter patients are at higher risk for serious COVID-19, early recognition is important to provide supportive care.

## Materials and methods

### Study design, sample size, and participants

Whilst it was not possible to calculate an appropriate sample size due to the rapid escalation of the COVID-19 pandemic and concomitant resource constraints experienced by the study centres during the time of planned data collection, the study team set a minimum target of 500 patients, of which at least 250 were admitted to intensive care, and that the study would remain open for enrolment of patients until 6 April 2020 to increase the patient numbers as much as possible.

In this explorative multicentre study patients were prospectively enrolled into a prognostic score development cohort from 21 February to 6 April 2020, with follow-up to document clinical outcome until 9 June 2020, from seven hospitals in the Netherlands and one each in Italy and Belgium.

Data analysis and prognostic score development was performed retrospectively from 9 to 29 June 2020. Thereafter, eligible patients presenting between 7 April to 15 May 2020 and 3 March and 17 May 2020 respectively at two hospitals in the Netherlands were retrospectively enrolled into a validation cohort.

Inclusion criteria for both cohorts were: primary presentation at participating hospitals; RT-PCR confirmed COVID-19; age ≥18 years;≥1 CBC-DIFF (±reticulocyte measurement (RET)) analysed on a Sysmex XN series haematology analyser (Kobe, Japan) as part of routine care; initial management decision record (self-isolation, general ward admission, ICU admission). Any documented comorbidities, duration of symptoms, subsequent ICU transfer, and outcome (recovered, died, unknown) were recorded.

Patients younger than 18 years were excluded as ethical approval was limited to adult subjects. Patients with an unknown outcome were excluded from analysis. Also, patients with a known underlying condition, such as a haematological malignancy or disorder, or concurrent chemotherapy, were excluded as such conditions are associated with abnormal blood cell counts which may either mask the severity of disease or falsely amplify it.

All CBC-DIFFs (+/- RET) done at presentation and throughout hospitalisation, of eligible patients, were included for retrospective analysis. All patients that met the inclusion criteria were included in the longitudinal haemocytometric data trend analysis (secondary aim) whereas only those patients that had at least one haematology profile measurement available on one or more days, on day 0, 1, 2, or 3 were included in the prognostic score development (primary aim).

Due to the rapidly changing dynamics of the clinical course of COVID-19, and the timing of blood testing being entirely at the discretion of the clinician, if a patient had more than one measurement

on a single day all measurements were included in the analysis as our objective is to have a prognostic score that is universally applicable to measurements taken at any time point between days 0-3.

Reference data were obtained from a random sample of 12,782 healthy individuals from a large population-based cohort in the Netherlands (http://www.lifelines.nl).

## Haemocytometry

Parameters included in this study are shown in *Table 7*.

In brief, standard CBC-DIFF parameters plus nucleated red blood cells (NRBC), immature granulocytes (IG), neutrophil reactivity index (NEUT-RI), and neutrophil granularity index (NEUT-GI) were measured. Neutrophil-to-lymphocyte ratio (NLR) and immature granulocyte-to-lymphocyte ratio (IGLR) were calculated.

Where available, we assessed reticulocyte count (RET), reticulocyte haemoglobin content (RET-He, the optical measurement of reticulocyte haemoglobinisation), the difference in reticulocyte and RBC haemoglobin content (Delta-He) (which is also a real time marker for iron bioavailability, with a negative value being an indirect marker of monocyte activation), the percentage of hypochromic (HYPO-He), and microcytic RBC (MICRO-R).

Anonymous XN-analyser raw data files were provided to Sysmex Europe collaborators for analysis using virtual analyser software, specifically to obtain reactive lymphocyte (RE-LYMPH), antibody-synthesising lymphocyte (AS-LYMPH), and reactive monocyte (RE-MONO) counts. If the analyser was equipped with a PLT-F (optical platelet count) measurement channel, and the initial CBC platelet count result triggered a reflex PLT-F measurement, or this was included in the initial request as a default profile, then immature platelet fraction (IPF#, IPF%) values measured in this channel were included in the data analysis. However, as most analysers in this study were not equipped with the PLT-F channel, the IPF# and IPF(%) values were partly derived from the RET channel measurement using virtual analyser software.

## Definitions

Disease severity was scored as follows: *Mild*: no hospitalisation, recovered; *Moderate*: ≤5 days hospitalisation without ICU/ventilation, recovered; *Severe*: >5 days hospitalisation without ICU/ventilation, recovered; *Critical*: ICU/ventilation at any stage of hospitalisation, recovered; *Fatal*: death.

Outcomes were classified as either recovered, died or unknown. 'Recovered' refers to all patients that had shown significant improvement and/or resolution of their clinical symptoms and thus discharged from hospital at the discretion of the attending physician. All patients that were never hospitalised (mild disease classification) were assumed to have recovered.

'Died' refers to all patients that died in hospital. Patients who were still hospitalised at the time of study termination or those that had been transferred to another healthcare facility and who's outcome could not be retrieved, were classified as having an 'unknown' outcome. ICU admission, whether direct or a subsequent transfer from the general ward, was not considered an outcome, but rather a criterion to define critical illness.

Two disease severity groups were defined, assuming that all patients that died would have needed ICU admission, and would have been admitted to ICU, had an ICU bed been available: (1) non-critical illness (NC) group comprising patients classified as mild, moderate or severe, and (2) the critical illness (CI) group comprised of critical (ICU survivors) and all fatal outcome patients, irrespective of general ward or ICU admission.

Day 0 refers to the day of first presentation at hospital, and the day of admission for those patients requiring hospitalisation. Day 1 refers to one day after the day of admission, or alternatively, one hospital bed night. Day 2 refers to 2 days, and so on.

## Trend analysis of single parameters and development of a prognostic score to predict disease severity and progression in COVID-19 patients

All available haemocytometric measurements from days 0-13 were included in the individual haematology parameter longitudinal analysis. Haemocytometric data were grouped according to clinical severity, symptom duration and days of hospitalisation, and analysed up to day 14 and compared with healthy controls, to identify specific patterns and trends. Haemoglobin values were adjusted for age and gender.

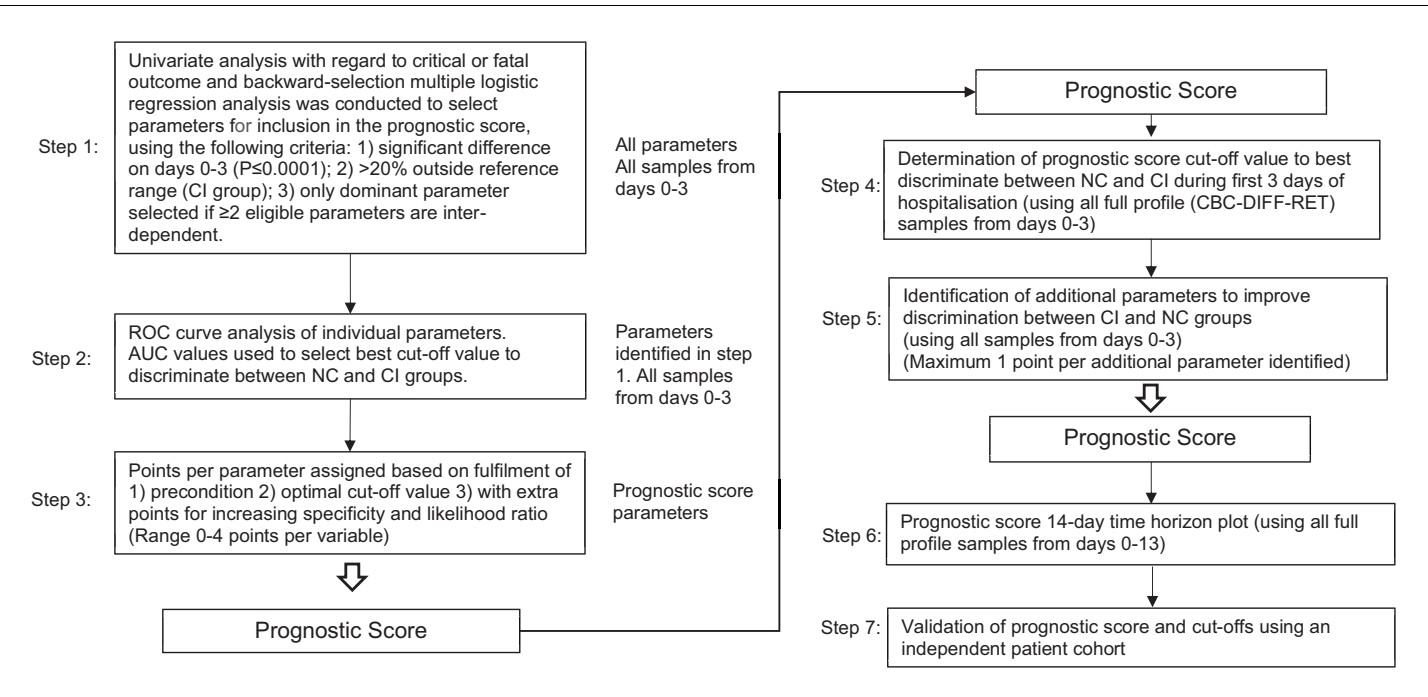

**Figure 12.** Flow chart illustrating steps involved in prognostic score development. Abbreviations: NC, non-critical patient group; CI, critical illness patient group; ROC, receiver operating characteristics (curve); AUC, area under the curve.

The prognostic score development process is outlined in *Figure 12*. The day 0 to 3 data points from the trend analysis were utilised for this purpose. In brief, each parameter was analysed univariately with regard to non-critical or critical illness classification and backward-selection multiple logistic regression analysis was conducted to select parameters for inclusion in the prognostic score, using the following criteria: (1) significant difference on days 0-3 (p≤0.001); (2)>20% abnormal in the CI group; (3) only dominant parameter selected if ≥2 eligible parameters are inter-dependent.

Receiver operating characteristics (ROC) curve analysis of selected parameters was applied for calculation of optimal cut-off values associated with sensitivity and specificity and used as the baseline for points assignment, with further increments for increasing specificity and likelihood ratio in discriminating between NC and CI groups (maximum four points). The score represents the sum of individual parameter points.

Score values were calculated for day 0-3 measurements to determine the cut-off value that best discriminate between the NC and CI groups. To further enhance score sensitivity without compromising specificity, additional parameters were included (maximum one point each) if they were significantly abnormal from day 4 onwards with >95% specificity.

The 14-day prognostic score time horizon was plotted using all available measurements and its performance in predicting disease severity validated in an independent patient cohort.

Only haemocytometry parameters were used as predictors in the prognostic score development. As these data are generated from automated haematology analysers, and do not rely on interpretation, predictors for the prognostic score were automatically blinded.

Also, the disease severity groups were defined in advance of patient enrolment, and patients were classified according to objective data (length of hospital stay, general ward or ICU, recovered and discharged from hospital or died) providing by the enrolling study centre prior to the commencement of data analysis and score development. In this regard, the authors involved in score developed had no influence over assessment of prediction of disease severity of individual patients, and hence assessment of outcome was deemed to have been blinded.

## Statistics

MedCalc Statistical Software version 19.2.1 (MedCalc Software Ltd, Ostend, Belgium; https://www.medcalc.org; 2020) was used. Differences between patient groups were assessed using Student's *t*-test for normality and if rejected (D'Agostino-Pearson test for normal distribution), the Mann-Whitney *U*-test for non-normally distributed variables was used. The Hodges-Lehmann median difference (md) was used to describe the differences in values of non-standard distributed parameters. To evaluate predictive values, we calculated the AUC of ROC curves, and 95% confidence intervals for each of the first 14 days of hospitalisation. This was done for both the development and validation cohort. The influence of two categorical variables was visualised using the clustered multiple comparison graph with standard error of mean (SEM). Confoundance of prognostic score, age, gender, and presence of comorbidities was tested by logistic regression. Box and whisker plots were used to visualise comparisons of multiple groups.

## Missing data

Those samples that did not have a RET channel measurement were excluded from the trend analysis of all parameters measured or derived from this measurement channel (*Table 7* ). Only samples with full profile (CBC-DIFF-RET) measurements were included for determination of cut-off values for prognostic score calculation. Only patients with full profile measurements were included in the validation cohort. No imputation was done for missing data. All patients that were enrolled from those centres that did not provide any information on the presence or absence of comorbidities, were excluded from the statistical analysis of prognostic score performance versus presence of comorbidities (classical risk factors) in predicting disease severity outcome.

# Acknowledgements

We would like to thank all patients involved in this study, as well as doctors, nurses, and researchers working together to fight against COVID-19. We also acknowledge Claudia Wienefoet of Sysmex Netherlands/Belgium for her support to the Dutch and Belgian participating centres and Sysmex Europe throughout the duration of this study. Sysmex Europe provided free of charge reagents to the study sites. No other funders were involved in this study.

# Additional information

### Competing interests

Joachim Linssen, Jarob Saker, Marion Münster: is a permanent employee of Sysmex Europe GMBH who provided free of charge study reagents to the study centres. Andre J van der Ven: has an ad hoc consultancy agreement with Sysmex Europe GMBH who provided free of charge study reagents to the study centres. The other authors declare that no competing interests exist.

### Funding

Sysmex Europe GMBH provided free of charge reagents for the study. No monetary payments were made to any of the investigators. Joachim Linssen, Jarob Saker and Marion Münster are full-time employees of Sysmex Europe GMBH and Andre van der Ven has an ad hoc consultancy agreement with Sysmex Europe GMBH.

### Author contributions

Joachim Linssen, Conceptualization, Data curation, Formal analysis, Funding acquisition, Validation, Visualization, Methodology, Writing - original draft, Writing - review and editing; Anthony Ermens, Conceptualization, Investigation, Writing - original draft, Writing - review and editing; Marvin Berrevoets, Michela Seghezzi, Giulia Previtali, Henk Russcher, Annelies Verbon, Judith Gillis, Jürgen Riedl, Eva de Jongh, Imke CA Munnix, Anthonius Dofferhof, Volkher Scharnhorst, Heidi Ammerlaan, Kathleen Deiteren, Stephan JL Bakker, Lucas Joost Van Pelt, Yvette Kluiters-de Hingh, Mathie PG Leers, Investigation, Writing - review and editing; Simone van der Sar-van der Brugge, Investigation; Jarob Saker, Data curation, Formal analysis, Validation, Visualization, Methodology, Writing - original

draft, Writing - review and editing; Marion Münster, Formal analysis, Validation, Visualization, Writing - original draft, Writing - review and editing; Andre J van der Ven, Conceptualization, Formal analysis, Validation, Visualization, Methodology, Writing - original draft, Project administration, Writing - review and editing

### Author ORCIDs
Simone van der Sar-van der Brugge ⬤ http://orcid.org/0000-0001-6462-075X
Lucas Joost Van Pelt ⬤ http://orcid.org/0000-0001-5538-1806
Andre J van der Ven ⬤ https://orcid.org/0000-0003-1833-3391

### Ethics
Human subjects: The study was reviewed by all participating centre ethics committees with approval granted in Italy (Registration Number 54/20) and Belgium (Registration Number 3002020000105) and exemption in the Netherlands, with need for informed consent waived by all.

### Decision letter and Author response
Decision letter https://doi.org/10.7554/eLife.63195.sa1
Author response https://doi.org/10.7554/eLife.63195.sa2

## Additional files

### Supplementary files
- Source data 1. all data file.
- Supplementary file 1. Basic demographic characteristics of COVID-19 PCR confirmed patients enrolled for prognostic score development demographic data by hospital.
- Supplementary file 2. Novel parameters of different manufacturers in relation to possible adaptability of the haemocytometric COVID-19 prognostic score.
- Transparent reporting form
- Reporting standard 1. TRIPOD checklist.

### Data availability
All data analysed during this study are included in the manuscript and supporting files. Source data files have been provided for figures 2, 3, 4, 5, 6, 7, 8, 9 and 11, and tables 1, 2, 4, 5, and 6.

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
