## [Decision Letter]

**Acceptance summary:**

This manuscript clearly highlights the value of haemocytometry in COVID-19 risk stratification and proposes a better way of using it.

**Decision letter after peer review:**

Thank you for submitting your article "A novel haemocytometric COVID-19 prognostic score developed and validated in a multicentre European hospital-based study" for consideration by *eLife*. Your article has been reviewed by three peer reviewers, including Anurag Agrawal as the Reviewing Editor and Reviewer #1, and the evaluation has been overseen by Mone Zaidi as the Senior Editor. The following individuals involved in review of your submission have agreed to reveal their identity: Minjie Chu (Reviewer #2); Chao Xu (Reviewer #3).

The reviewers have discussed the reviews with one another and the Reviewing Editor has drafted this decision to help you prepare a revised submission.

Summary:

The reviewers note that COVID-19 mortality is low but highly variable and not restricted to only the elderly. Thus prediction tools for high risk subjects, as described here, may be useful by permitting early institution of treatment and higher care level. While Neutrophil-lymphocyte ratio (NLR) is well established by now as a good predictor of mortality, other hemocytometric parameters may add additional value and that is investigated nicely here. The approach is plausible and was validated in retrospective multi-centric data, with a separate validation set. This is the strength of the paper

Essential revisions:

The major concerns relate to how longitudinal data was analysed and whether it was appropriate since there were insufficient details. Please consider each question from that standpoint and see if any new analysis is needed.

1) Heterogeneity of data from multiple centers? Sample size of each hospital? In addition to report the demographic info by groups, the demographic data among multiple centers should be compared.

2) Looks like some patients had multiple measurements on days 0-3, while some patients had only one measurement? How did the authors analyze the multiple measurements?

3) How did the authors test the group difference over time? Usually the linear or genialized linear mixed model should be used to analyze longitudinal data. However, there is no related description in the Materials and methods – Statistics section. For example, the p-value in Figure 4B.

4) Did the authors using all data from day 0-3 to develop a general model? But the model calculated the prognostic score using data from single day. If the authors used all data, what method was used to analyze the longitudinal data?

5) How were the points developed for different predictors? From the logistic regression coefficient?

6) How did the authors calculate the ROC over 14 days, when there were prognostic scores for each day? There are overlap samples over 14 days.

---

## [Author Response]

Essential revisions:The major concerns relate to how longitudinal data was analysed and whether it was appropriate since there were insufficient details. Please consider each question from that standpoint and see if any new analysis is needed.1) Heterogeneity of data from multiple centers? Sample size of each hospital? In addition to report the demographic info by groups, the demographic data among multiple centers should be compared.

We agree that the inclusion of the sample size and breakdown of patient demographic data per hospital will provide the reader insight into the heterogeneity of the data. We have included this information as Supplementary file 1. In brief, the sample size of each hospital ranged from 20 to 252; median age ranged from 62 to 75.5 years; male representation ranged from 61.2% to 76.5%; median length of hospitalisation was highly variable from 15.5 to 5 days (the smaller sample sizes had the highest median value); presence of comorbidities ranged from 53% to 75%; mortality ranged from 7% to 32%. Four hospitals did not include any mild cases and the proportion of critical illness (previously referred to as critical/fatal group) ranged from 36% to 56%.

2) Looks like some patients had multiple measurements on days 0-3, while some patients had only one measurement? How did the authors analyze the multiple measurements?

All multiple measurements were included in the analysis and treated equally. Prognostic score values were calculated for each available measurement per day. All patients classified as mild, by definition, were not hospitalised and therefore had only a single measurement on day 0, and no further measurements on days 1 to 3. If a hospitalised patient had a measurement on more than one day in the period day 0 to day 3, then that patient would be represented on each day for which there was a measurement available. If a patient had more than one measurement on a single day, then all measurements from that single day were included.

All measurements taken were at the discretion of the attending clinician. Repeat measurements on a single day may have occurred when the initial sample was taken in the emergency department and was subsequently repeated as part of the routine admission care when a patient entered the hospital ward/ICU or subsequent transfer from ward to ICU. Also, if a patient showed signs of deterioration, that is, a more critical course, the likelihood of multiple measurements is greater. This was borne out by the breakdown of occurrence of multiple single day measurements between the clinical severity groups. In the non-critical group, there were 5 patients with two measurements each on a single day, whereas in the critical/fatal group (now renamed “critical illness” group), there were 23 patients with multiple measurements as follows: 20 x 2 measurements, 2 x 3 measurements and 1 x 4 measurements. In total there were 35 occurrences of multiple measurements on a single day.

Overall, the occurrence of two or more daily measurements for individual patients was low (5%). Although the critical illness group had disproportionately more multiple measurements, we do not believe that this unduly influenced the overall observation that the prognostic score, in this group, increased over time. Upon review of the score values obtained for the two or more samples taken on the same day from individual patients with the critical illness, we observed that repeated measurement prognostic score results were either the same or higher in keeping with illness progression, and as such, exclusion of these repeated measurements would not have flattened the prognostic score curve. In contrast, the non-critical patient group all had a stable course with the prognostic score not increasing over time. Had any additional measurements been taken for individuals in the non-critical group, we assumed that they would have provided score values similar to both the previous measurement and the next measurement taken on a subsequent day.

Due to the rapidly changing dynamics of the clinical course of COVID-19, we believe that it is justified to include multiple measurements from a patient on a single day as we want the score to be universally applicable to measurements taken at any time point between days 0-3.

We have added the following paragraph to the Materials and methods section under study design, sample size and participants, to clarify that all multiple measurements were included.

**“**Due to the rapidly changing dynamics of the clinical course of COVID-19, and the timing of blood testing being entirely at the discretion of the clinician, if a patient had more than one measurement on a single day all measurements were included in the analysis as our objective is to have a prognostic score that is universally applicable to measurements taken at any time point between days 0-3.”

3) How did the authors test the group difference over time? Usually the linear or genialized linear mixed model should be used to analyze longitudinal data. However, there is no related description in the Materials and methods – Statistics section. For example, the p-value in Figure 4B.

We agree with the reviewer that when comparing two groups for whom longitudinal data has been plotted, that one would ordinarily expect to also see statistical analysis of group differences over time. We however did not do this analysis for the following reasons:

1) The patient group is inherently heterogeneous as health-seeking behaviour may vary widely between patients: some patients may present early while others present late at health facilities. Furthermore, recall when symptoms started may be unclear at time of presentation to the hospital. Therefore, a patient on day 3 of hospitalisation could be in an earlier phase of infection than a patient on day 0.

2) The intended use of the prognostic score was to identify at an early stage (once they had sought medical care) who is likely to deteriorate: to keep it as simple and practical as possible, we analysed the data from day 0-3 as a collective dataset for the purpose of identifying which parameters, and at what cut-off values, should be incorporated in the score. We did not compare score values from the early phase of hospitalisation (days 0-3) with the later phase (days 4-13) as the intention was to predict outcome early on before signs of critical illness may become clinically overt.

We did however plot the prognostic score data over time to document the patterns as this may be useful in the future to assess response to specific COVID-19 therapies.

Figure 3B is a comparative plot of NLR values for the non-critical patient group and critical illness patient group over the first 14 days of hospitalisation. Figures 3 to 7 support the secondary aim of the study which is to document trends of haematology parameters over time, specifically for the newer parameters, as most of the published data in COVID-19 patients focussed on traditional parameters such as lymphocyte counts, platelet counts and NLR. In this regard we also did not conduct any statistical analysis for the longitudinal data of the individual haematology parameters. The statistical analysis of group differences for the day 0-3 data, which is the baseline data that served as the starting point of our prognostic score development, was done for all parameters and is shown in Table 3. Comparison of median NLR values for the non-critical and critical illness groups gave a P value of <0.0001.

We have not made any mention in the Materials and methods of the revised manuscript as to why longitudinal data analysis was not done but would be happy to do so if the editors believe that it is needed.

We have added the following paragraph to the Discussion to address the reason for the absence of longitudinal data analysis.

“The patient group is inherently heterogeneous as health-seeking behaviour may vary widely between patients: some patients may present early while others present late at health facilities. […] We did however plot the data over time to document the patterns as this may be useful in the future to assess response to specific COVID-19 therapies.”

4) Did the authors using all data from day 0-3 to develop a general model? But the model calculated the prognostic score using data from single day. If the authors used all data, what method was used to analyze the longitudinal data?

The response provided to point #3 above is also relevant to this reviewer query. The patient group was heterogenous with respect to duration of symptoms at the time of hospital presentation, with the 95% CI on day 0 ranging from 1-20 days, and 4-21 days on day 3. Based on this observed, and expected, heterogeneity, and our aim, which was to develop a prognostic score to serve as a risk stratification tool for the early identification of patients likely to have a critical clinical course, we pooled all data from days 0-3 to develop the prognostic score. Pooling data from days 0 to 3 would ensure that the score is representative and usable for any single measurement taken on days 0 to 3.

Score development entailed the selection of individual haematology parameters with the highest discriminatory power between the two clinical severity groups, as well as the assignment of quantitative cut-off values for each parameter. To demonstrate the feasibility of correctly identifying disease severity early on (clinical applicability) in individual patients, independent of which day (0-3) the sample was taken, score values were calculated per single measurement and plotted per single day by group (NC or CI).

The aim of the score is to predict early on, using a single haematology profile measurement, if a patient is likely to develop critical illness, before symptoms of critical illness may become overt. In this regard, we did not conduct a longitudinal data analysis of prognostic score values as we believe that the clinical applicability of our prognostic score as a predicate tool for the identification of “high risk” patients, with the purpose of supporting the initiation of timely interventions, would become diminished, if not obsolete, as after several days of hospitalisation, symptoms in keeping with a worsening condition are very likely to have manifested clinically.

5) How were the points developed for different predictors? From the logistic regression coefficient?

The allocation of score values was defined in the footnote of Table 4, the details of which were as follows:

“Note: For the primary variables, 1 point = value above the cut-off value for the best AUC; 2 points = value above the cut-off value for the best AUC and ≥80% specificity; 3 points = value above the cut-off value for the best AUC and >90% specificity; and 4 points = value above the cut-off value for the best AUC and >95% specificity. The cut-off values for the secondary variables were chosen exclusively based on observed extremes of values in critical disease, with the maximum award of 1 point per variable.”

6) How did the authors calculate the ROC over 14 days, when there were prognostic scores for each day? There are overlap samples over 14 days.

We agree with the authors that it seems counterintuitive to calculate a ROC curve over a 14-day period, when scores from individual days had to be considered. In this regard, the AUCs as shown in Figure 9C are exclusively intended to demonstrate that in our study the overall performance of the prognostic score was better than other previously published single prognostic parameters, such as NLR. Data was taken over the entire 14-day period for all parameters (prognostic score, NLR, absolute lymphocyte count, absolute monocyte count, platelet count and platelet-to-lymphocyte ratio) and measurements for each were only included if they were available for all parameters shown in the graph. We agree that if we were to do a standalone ROC curve analysis exclusively for prognostic score performance, then we would calculate the ROC curve using only a single measurement per patient, i.e. the highest score value obtained per patient on either days 0, 1, 2 or 3.

AUC values for individual days are shown in Table 6.